# `TelecomTS`: A Multi-Modal Observability Dataset for Time Series and Language Analysis

Austin Feng [* 1]  Andreas Varvarigos [* 1]  Ioannis Panitsas [1]  Daniela Fernandez [1]  Jinbiao Wei [1]  Yuwei Guo [2]
Jialin Chen [1]  Ali Maatouk [1]  Leandros Tassiulas [1]  Rex Ying [1]

## Abstract

Modern enterprises generate vast streams of time series metrics when monitoring complex systems, known as observability data. Unlike conventional time series from domains such as climate, observability data are zero-inflated, highly stochastic, and exhibit minimal temporal structure. Despite their importance, observability datasets remain underrepresented in public benchmarks due to proprietary restrictions and privacy concerns. Existing datasets are often anonymized and normalized, removing scale information and limiting their use for tasks such as anomaly detection, root cause analysis, and multi-modal reasoning. To address this gap, we introduce `TelecomTS`, a large-scale observability dataset derived from a 5G telecommunications network. `TelecomTS` features heterogeneous, de-anonymized covariates with explicit absolute scale information and provides a diverse suite of downstream tasks, including anomaly detection, root cause analysis, and multi-modal question-answering. Benchmarking state-of-the-art time series, language, reasoning, and multi-modal foundation models reveals that existing approaches struggle with the abrupt, noisy, and high-variance dynamics characteristic of observability data. Our experiments further underscore the importance of preserving covariates' absolute scale, emphasizing the need for foundation time series models that natively leverage scale information for practical real-world observability applications. The code is available at: `https://github.com/Ali-maatouk/TelecomTS`.

## 1. Introduction

Time series data is ubiquitous across fields such as weather, finance, and energy systems (Hu et al., 2025; Kong et al., 2025a; Farahani et al., 2023; Noshad et al., 2019; Fassois & Sakellariou, 2009). One particular domain that has been under-studied but is now garnering increasing attention is the observability domain, which analyzes time series metrics generated by monitoring complex systems to detect anomalies, diagnose issues, and maintain system health (Cohen et al., 2025; Palaskar et al., 2024). This observability data includes CPU and memory utilization, network throughput, request latency, error rates, and disk I/O, each offering critical insight into the state and performance of the system.

Compared to data found in climate or other commonly studied time series domains, observability data is fundamentally different and poses unique modeling challenges due to its distinctive characteristics. First, it is highly zero-inflated: many metrics track infrequent events, such as bursts of user traffic, resulting in sparse time series dominated by zeros and punctuated by informative spikes. Second, it displays highly dynamic patterns characterized by frequent, abrupt transitions that are challenging to model (Datadog, 2024). Finally, observability data is highly stochastic, with metrics often appearing irregular and exhibiting minimal discernible temporal structure (Cohen et al., 2025).

Despite their importance and the challenges they present, these types of time series data remain relatively understudied in the time series literature. This gap can be attributed to several factors: (1) the lack of publicly available datasets due to the proprietary nature of observability data, (2) anonymization in the few publicly available datasets, which obscures both the identity of the metrics and vital information such as their absolute scale; and (3) the scarcity of downstream tasks, such as anomaly detection, root cause analysis, and multi-modal reasoning, in existing publicly available observability datasets, despite their critical importance in the observability domain.

Our paper aims to bridge this gap by introducing `TelecomTS`, a large-scale observability dataset focused on the telecommunications domain. An overview of

---
[*]Equal contribution  [1]Yale University, New Haven, CT, USA [2]Johns Hopkins University, Baltimore, MD, USA. Correspondence to: Ali Maatouk <ali.maatouk@yale.edu>.

*Proceedings of the 43rd International Conference on Machine Learning*, Seoul, South Korea. PMLR 306, 2026. Copyright 2026 by the author(s).

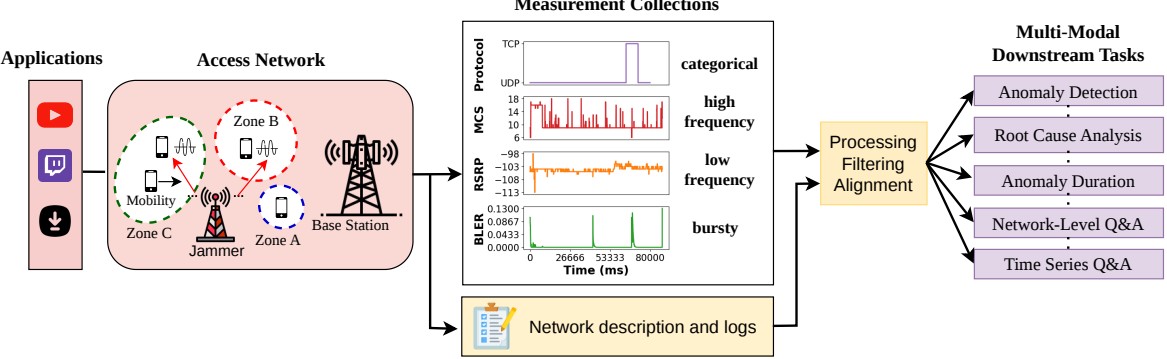

*Figure 1.* An overview of `TelecomTS`, illustrating its data curation pipeline, covariate characteristics, and the range of supported multi-modal downstream tasks.

`TelecomTS` can be found in Fig. 1. Compared to prior datasets, `TelecomTS` differs in two major ways:

**1. Heterogeneous, de-anonymized covariates with scale information:** Built from extensive data collection across a 5G network, `TelecomTS` contains millions of observations of key performance indicators (KPIs). Moreover, the dataset captures categorical covariates from dynamically changing communication protocols along with mixed-type metrics (integers and floating-point variables with diverse ranges and distinct statistical distributions), thus reflecting the inherently heterogeneous nature of observability data. Crucially, it provides full visibility into the absolute scale of each covariate, thereby enabling systematic investigation of how scale information and normalization strategies affect downstream task performance in observability settings.

**2. Comprehensive suite of downstream tasks:** Since observability applications extend beyond forecasting, `TelecomTS` is designed to support a broad suite of multi-modal downstream tasks. Particularly, our dataset incorporates a diverse spectrum of anomalies, including real anomalies generated via controlled jamming signals as well as synthetically curated rare events grounded in scholarly accounts of real-world system failures. These anomalies naturally give rise to tasks such as anomaly detection, anomaly duration estimation, and root cause analysis, all of which we uniformly instantiate as question-answering (Q&A) problems. In addition, the dataset includes Q&A instances that require temporal reasoning over time series dynamics alongside network-specific queries that reflect the semantics of the observability environment. Finally, motivated by the growing importance of reinforcement learning for reasoning in time series Q&A settings (Zhang et al., 2025; Parker et al., 2025), we annotate each question–answer pair with explicit reasoning paths, providing grounded justifications that support reasoning-aware training and evaluation.

By benchmarking time series models, language models, rea-soning models, and multi-modal models on `TelecomTS`, we show that state-of-the-art approaches consistently struggle with the abrupt, noisy, and high-variance dynamics characteristic of observability data. These challenges manifest as elevated false-positive rates in anomaly detection, misidentified root causes, and degraded performance on time series question-answering tasks. Moreover, our experiments highlight the pivotal role of preserving covariates' absolute scale in improving downstream task performance, underscoring the need for time series models that explicitly account for scale information to achieve robust performance in real-world observability applications.

## 2. Related Works

**Time Series Foundation Models**. Recent advances in time series foundation models (Ansari et al., 2024; Woo et al., 2024; Das et al., 2024) have demonstrated strong zero-shot performance on time series benchmarks such as GIFT-EVAL (Aksu et al., 2024). Trained on large, multi-domain time series corpora, these models have emerged as a dominant paradigm for time series learning, as they support zero-shot inference and require minimal fine-tuning when adapting to downstream tasks (Kottapalli et al., 2025; Faw et al., 2025).

**Time Series Datasets**. The datasets used to train these foundation models span a wide range of domains, including energy (Zhou et al., 2021), climate (Mouatadid et al., 2024), and sales (Makridakis et al., 2022; Jiang et al., 2024). In addition, meta-datasets that aggregate multiple sources have been introduced, most notably Monash (Godahewa et al., 2021) and the Time Series Pile (Goswami et al., 2024). Despite their broad coverage, these datasets largely exclude observability data, which remains scarce in the literature due to its proprietary nature (e.g., customer traffic data from a cloud operator) (Xie et al., 2025; Qureshi et al., 2023).

**Observability Datasets**. Given this gap, time series founda-

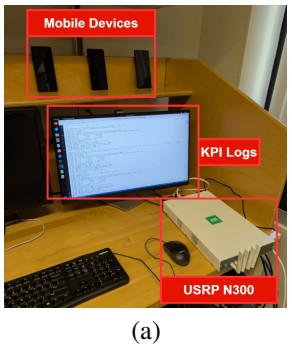 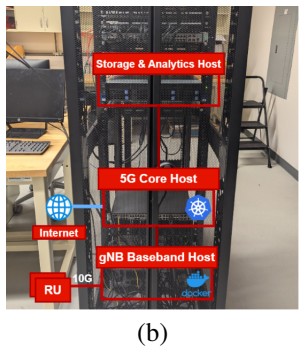 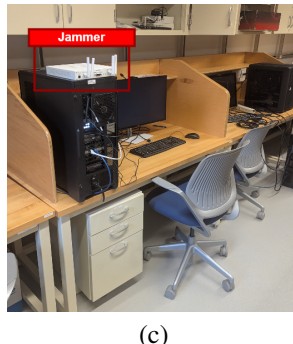

(a)  (b)  (c)

*Figure 2.* Overview of the 5G wireless network for data collection: (a) mobile devices generating network traffic; (b) server infrastructure hosting the core network and base-station workloads; (c) programmable jammer introducing controlled over-the-air interference.

tion models have been shown to underperform on observability data (Toner et al., 2025; Palaskar et al., 2024), motivating growing community efforts to bridge the gap by curating and publishing observability-focused datasets. A notable recent contribution in this direction is the BOOM dataset (Datadog, 2024; Cohen et al., 2025), which consists of real-world metrics collected from Datadog. BOOM captures a wide spectrum of observability signals from distributed systems, including infrastructure, database, and security.

**Lingering Gaps**. Despite the advancements introduced by the BOOM dataset, several limitations remain. First, the data is anonymized, providing no information about the time series variates. Second, the dataset consists of time series observations that are normalized to preserve privacy. These constraints have multiple consequences: (1) anonymization limits the ability to augment these time series observations with downstream tasks such as anomaly detection, multimodal reasoning, and question-answering as has been done in other domains like finance and weather (Dau et al., 2018; Liu et al., 2024a; Chen et al., 2025), since the identities of the covariates are obscured; (2) normalization and loss of absolute scale obscure critical information, particularly in observability contexts where metric magnitudes (e.g., CPU load) are essential for downstream tasks like anomaly detection (Lin et al., 2024); and (3) BOOM focuses solely on numerical time series, providing no support for tasks that combine time series data with natural language. Consequently, there remains a strong need for de-anonymized observability datasets that provide fully detailed metrics and support a broad range of multi-modal downstream tasks.

## 3. TelecomTS Dataset

In this section, we provide a detailed overview of the curation process for TelecomTS, including the time series observations, anomalies, and question–answer pairs. A detailed comparison of the covariate behaviors in TelecomTS versus those commonly found in the literature is provided in Appendix A.

### 3.1. Raw Data Collection

**5G Network**. Since telecommunications data is usually proprietary to network operators, comprehensive open-source datasets in this field remain limited. For this reason, we collect our networking data using a 5G wireless network developed in our lab, free of privacy concerns. The setup consisted of a single base station (gNB) connected to a full-stack 5G core network serving as the Internet gateway. A mobile device was connected to the network and used to generate live traffic using real-world applications such as YouTube, Twitch, and file downloads. To ensure the collected data captures a diverse range of operating conditions, we deliberately varied network configurations, transmission power levels, traffic patterns, and congestion levels across collection sessions (see Appendix B). This, combined with the inherent phenomena of our 5G radio environment such as interference, congestion, and scheduling dynamics, supports generalizability of our dataset beyond the testbed setting. The overall architecture of the network is illustrated in Fig. 2(a) and Fig. 2(b).

**Measurements Collection**. During each user connection to the internet, 18 KPIs were recorded from both the base station and the device at 100 ms resolution. Since the data was collected across two separate traces, each capturing different types of KPIs, a time misalignment offset was introduced between the traces. To correct for this offset, we selected two highly correlated KPIs from each trace and applied a histogram-matching technique. Specifically, we aligned the two traces by temporally shifting one relative to the other and finding the time offset that minimizes the Kullback–Leibler (KL) divergence between their histograms (see Appendix B.2 for details).

**Zoning for Radio Conditions**. To introduce spatial variability in our data, the lab environment was divided into three zones based on distance from the base station. Zone A (0–3 m) provided strong signal quality; Zone B (3–6 m) reflected moderate signal quality, and Zone C (>6 m) delivered

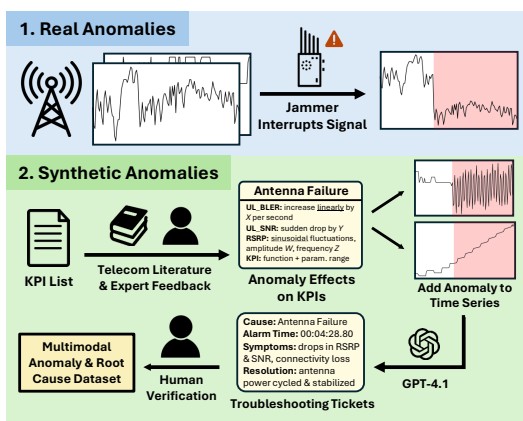

*Figure 3.* An overview of the anomalies curation process.

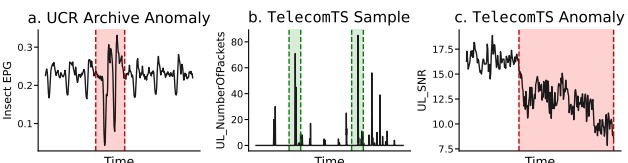

*Figure 4.* An illustrative difference between UCR Archive Anomaly dataset and the anomalies found in TelecomTS. The anomalies found in the former typically manifest as a clear deviation from an otherwise smooth and predictable trend.

weak signal environments.

**Mobility and Congestion**. Data was also collected under static and mobile conditions to reflect realistic user mobility. Congestion scenarios were emulated by introducing secondary devices that generated heavy traffic, creating resource contention representative of high-load environments.

Additional information on the network setup, collected KPIs, and layout is available in Appendix B.

### 3.2. Anomalies Curation

Popular anomaly detection datasets (e.g., UCR (Dau et al., 2018)) often contain both real and synthetic anomalies, as real anomalies are inherently rare and difficult to capture at scale. Following a similar methodology, TelecomTS integrates both real anomalies and a principled approach for generating synthetic ones, as shown in Fig. 3.

**Real anomalies.** In our setup, an adversarial jammer is employed to emit electromagnetic signals on the same frequencies used by mobile devices, thereby interfering with their transmissions. The jammer alternates between active and idle periods; during its active phase, it disrupts network communication, causing packet loss and anomalous behavior throughout the network. A visual illustration of this setup is shown in Fig. 2(c), and additional details on the jamming configurations are provided in Appendix B.2.

**Synthetic Anomalies**. To expand the coverage to the diverse range of fault scenarios encountered in operational networks, we extend our dataset with synthetic anomalies designed to replicate characteristic behaviors reported in the telecommunications literature. A key challenge in generating synthetic anomalies is ensuring that they faithfully mimic the characteristics of rare real-world network anomalies rather than introducing random drops or unsubstantiated perturbations in the time series. To address this, we adopt

a principled methodology for realistic synthetic anomaly creation, outlined as follows.

First, we curate a list of ten anomaly types known to occur in networked systems, drawing from technical manuals and scholarly material (Liu et al., 2023; Yen et al., 2022; Hasan et al., 2024). For each anomaly type, we identify the corresponding effect on KPIs present in our time series and validate these mappings against reported behaviors in prior work to ensure their relevance. For example, co-channel interference is modeled to degrade SNR, BLER, and PRB utilization as reported in (Tusha et al., 2024); congestion and buffer overflow exhibit PRB saturation and declining throughput (Andras et al., 2023); Doppler-induced channel aging leads to lower MCS and elevated BLER (Díaz-Ruiz et al., 2025); and ping-pong handover results in oscillatory signal quality and increased packet loss (Tsai et al., 2015).

Once anomalies and their symptoms are defined, we model their occurrence and duration. Following findings from large-scale operational networks (Maatouk et al., 2024), we model anomaly durations and inter-arrival times as exponentially distributed variables with empirically motivated rates. Using these models, we generate synthetic anomalies by manipulating the appropriate subset of covariates for each anomaly type in the raw measurements dataset. Full details are provided in Appendix C.

Finally, for each anomalous sample, we generate a textual troubleshooting ticket that serves as the foundation for reasoning trace synthesis (detailed in the next subsection). Each ticket specifies the anomaly type, start and end times, and provides a narrative description of the observed KPI behavior and symptoms during the event. We generate these tickets using GPT-4.1, conditioning on the selected anomaly type, associated symptoms, and temporal boundaries. All tickets undergo subsequent review and validation before being used for reasoning trace synthesis. The complete set of prompts used for this generation, along with all other prompts employed in this work, is included in Appendix D.

**A comparative example**. A key question that arises is how the anomalies in our dataset differ from those found in standard anomaly detection benchmarks such as the UCR archive. To address this, we provide a comparative visualiza-

*Table 1.* Statistical summary of `TelecomTS`.

| Statistic | Description | Count |
|---|---|---|
| **Time Series Samples** | Total samples | 32,000 |
| | Sample length | 128 |
| **Channels** | Total Channels | 18 |
| | Channel Types | 10 float, 6 integer, 2 categorical |
| **Anomalies** | Anomaly Types | 11 |
| **Q&A Categories** | Time Series Q&A Categories | 64 |
| | Network-Level Q&A Categories | 4 |
| | Anomalies Q&A Categories | 3 |
| **Total QA Size** | Total QA Instances | 2,210,185 |

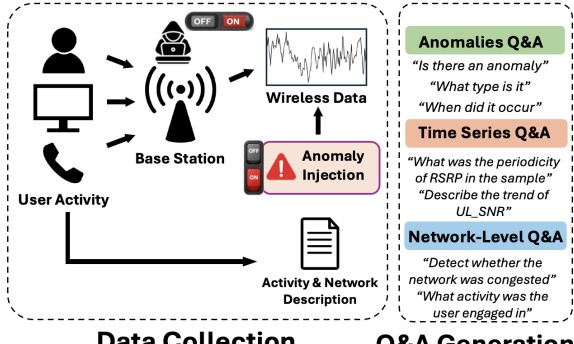

*Figure 5.* An overview of our Q&A pairs.

tion in Figure 4. Figure 4a displays a sample from the UCR anomaly dataset, where anomalies may appear as slight deviations from otherwise smooth trends. In contrast, Figure 4b presents a burst in user traffic found in our dataset, which is an abrupt yet entirely normal behavior of observability data. Figure 4c presents a true anomaly in our setting, where a sustained shift in the overall trend indicates a true fault rather than fluctuations typical of observability operations. This comparison highlights a fundamental distinction in our dataset: abrupt changes are often inherent to the system and do not necessarily signal anomalous behavior.

### 3.3. Questions and Answers Curation

Given the collected KPI measurements under both normal operation and anomalous conditions, we construct fixed-length time series samples by applying a sliding window of length 128 time steps with a stride of 32. This yields 32K samples, each consisting of 128 time steps across 18 KPI channels. Each time series sample is associated with structured metadata collected alongside the data, including network conditions, the presence or absence of anomalies, and their corresponding types.

With these samples and metadata available, we proceed with curation of question-answer pairs across three categories: (1) *anomalies Q&A*, (2) *time series Q&A*, and (3) *network Q&A*, along with explicit reasoning paths that provide grounded justifications for the answers in each category. These categories are designed to span a spectrum of difficulty, from statistical extraction over individual KPIs to cross-channel reasoning over coordinated changes across multiple signals. All dataset statistics are summarized in Table 1, and representative examples are shown in Fig. 5 and Appendix E.

**1. Anomalies Q&A.** This family of Q&A is directly derived from the anomaly annotations of each sample and includes questions about whether an anomaly is present, as well as its duration and root cause when one exists.

**2. Time Series Q&A.** This second family of Q&A targets time series reasoning and probes a model's understanding of intrinsic statistical and structural properties of the sig-

nals, such as mean, variance, periodicity, and trend. To that end, for each time series sample and KPI channel, we compute basic statistics such as mean and variance directly from the data. Periodicity is estimated using a Fourier transform, where the dominant frequency component determines the primary period. Trends are identified by fitting a linear regression line to each series and evaluating its slope: slopes above (below) one standard deviation from the mean across all samples are labeled as positive (negative), while remaining cases are treated as having no prominent trend.

**3. Network-Level Q&A.** The final family of Q&A focuses on network-level aspects. Specifically, from each sample's metadata, we extract labels corresponding to user activity, mobility state, zone, and network congestion status. We then generate natural-language questions about these specific labels, sampling from a diverse set of question templates to create a balanced Q&A set.

**Reasoning Traces.** Finally, for all network-level and anomaly Q&A instances, we curate explicit reasoning paths that provide grounded justifications for the answers. We focus on these categories as they require cross-channel reasoning over coordinated changes across multiple KPIs — for example, diagnosing a fault requires recognizing that BLER increases, MCS drops, and PRB spikes are co-occurring and interdependent — whereas time series Q&A targets statistical extraction from individual channels and does not require multi-step reasoning. These traces are designed to support both training and evaluation of reasoning-capable models on our dataset. To create these paths, for each selected instance, we generate reasoning traces using a large language model with rejection sampling over multiple candidate generations. To ensure faithfulness and reduce hallucinations, the model is conditioned on relevant metadata during trace generation, such as the troubleshooting tickets in the anomalies Q&A case. Afterwards, a scoring function is used to select high-quality traces based on structural coherence, conciseness, appropriate use of KPI references, avoidance of metadata leakage, and consistency between the reasoning process and

*Table 2.* Anomaly detection precision, recall, and F1 score.

| Model | Precision | Recall | F1 |
|---|---|---|---|
| *Large language models* | | | |
| GPT-4.1 (without context) | 0.200 | 1.000 | 0.333 |
| GPT-4.1 (with context) | 0.173 | 0.609 | 0.270 |
| Claude 3.7 Sonnet (without context) | 0.182 | 0.840 | 0.299 |
| Claude 3.7 Sonnet (with context) | 0.194 | 0.860 | 0.322 |
| *Reasoning models* | | | |
| o4-mini (without context) | 0.188 | 1.000 | 0.316 |
| o4-mini (with context) | 0.246 | 0.580 | 0.345 |
| DeepSeek-R1 (without context) | 0.259 | 0.600 | 0.362 |
| DeepSeek-R1 (with context) | 0.244 | 0.470 | 0.321 |
| *Foundation Models* | | | |
| Moment | 0.256 | 0.888 | 0.397 |
| Moirai2 | 0.346 | 0.490 | 0.405 |
| Toto | 0.521 | 0.750 | 0.615 |
| *Time series models* | | | |
| Mantis | 0.800 | 0.800 | **0.800** |
| Mantis (w/o scaling) | 0.585 | 0.850 | 0.692 |
| TimesNet | 0.389 | 0.652 | 0.487 |
| Autoformer | 0.199 | 0.690 | 0.308 |
| Non-stat. Transformer | 0.446 | 0.464 | 0.455 |
| FEDformer | 0.224 | 0.560 | 0.320 |
| Informer | 0.459 | 0.448 | 0.453 |
| *Multi-modal models* | | | |
| Toto+Qwen-3-4B | 0.368 | 0.717 | 0.487 |
| Toto+Qwen-3-4B+Thinking | 0.354 | 0.699 | 0.469 |

*Table 3.* Anomaly duration analysis precision, recall, and F1 score.

| Model | Precision | Recall | F1 |
|---|---|---|---|
| *Large language models* | | | |
| GPT-4.1 | 0.715 | 0.334 | 0.456 |
| Claude 3.7 Sonnet | 0.697 | 0.292 | 0.412 |
| *Reasoning models* | | | |
| o4-mini | 0.683 | 0.241 | 0.356 |
| DeepSeek-R1 | 0.641 | 0.349 | 0.448 |
| *Foundation Models* | | | |
| Moment | 0.556 | 0.940 | 0.699 |
| Moirai2 | 0.681 | 0.923 | 0.784 |
| Toto | 0.910 | 0.931 | **0.921** |
| *Time series models* | | | |
| Mantis | 0.873 | 0.914 | 0.893 |
| Mantis (w/o scaling) | 0.803 | 0.970 | 0.879 |
| TimesNet | 0.745 | 0.866 | 0.801 |
| Autoformer | 0.661 | 0.847 | 0.742 |
| Non-stat. Transformer | 0.667 | 0.828 | 0.739 |
| FEDformer | 0.661 | 0.853 | 0.745 |
| Informer | 0.660 | 0.854 | 0.744 |
| *Multi-modal models* | | | |
| Toto+Qwen-3-4B | 0.788 | 0.848 | 0.817 |
| Toto+Qwen-3-4B+Thinking | 0.690 | 0.716 | 0.702 |

the final answer. Additional details on the reasoning trace generation pipeline and quality assessment procedures are provided in Appendix F.

## 4. Experiments

In this section, we evaluate state-of-the-art models on the downstream tasks defined in TelecomTS. Our benchmark includes LLMs, reasoning models, time series models, and multi-modal models. Through these experiments, we reveal the performance gap that emerges when existing models are exposed to the complex nature of observability data captured by our dataset. Training details are provided in Appendix G.

### 4.1. Anomaly Detection

For this task, we evaluate each model on a randomly selected subset of 1,000 samples from our dataset, balancing computational cost across multiple large models while maintaining representative and stable evaluation. Results were adapted to ensure a 80%–20% ratio of normal to anomalous instances, consistent with anomaly prevalence ranges used in established benchmarks (Han et al., 2022; Xu et al., 2022). For language and reasoning models, we evaluate two settings: (i) without context, where the model directly judges anomaly presence from the time series, and (ii) with con-

text, where the model is informed that the data is naturally erratic and can have ups and downs as an intrinsic behavior. For time series models, we evaluate both foundation and standard architectures; in foundation models, the backbone is frozen and only a classification head is trained. For multi-modal models, we evaluate a Toto+Qwen-3-4B multi-modal time series LLM following an early-fusion design (Team, 2025), trained using LoRA.

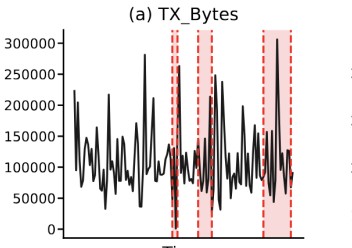 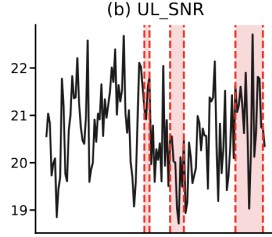

*Figure 6.* Illustration of a failure case that affected all benchmarked models on this specific sample.

**Results Analysis.** As shown in Table 2, models like GPT-4.1 and o4-mini exhibit a strong bias toward false positives (i.e., predicting normal samples as anomalous) when no contextual information is provided. Other language models display similar tendencies, though the bias is a bit less severe. This behavior stems from the inherent characteristics of our dataset: abrupt fluctuations are common in observational data, leading models to misinterpret erratic but normal behavior as anomalous. To illustrate this challenge, we pro-

*Table 4.* Accuracy in root cause analysis.

| Model | Acc |
|---|---|
| *Large language models* | |
| GPT-4.1 (without context) | 0.215 |
| GPT-4.1 (with context) | 0.227 |
| Claude 3.7-Sonnet (without context) | 0.115 |
| Claude 3.7-Sonnet (with context) | 0.245 |
| *Reasoning models* | |
| o4-mini (without context) | 0.245 |
| o4-mini (with context) | 0.275 |
| DeepSeek-R1 (without context) | 0.145 |
| DeepSeek-R1 (with context) | 0.261 |
| *Foundation Models* | |
| Moment | 0.550 |
| Moirai2 | 0.225 |
| Toto | **0.848** |
| *Time series models* | |
| Mantis | 0.590 |
| Mantis (w/o scaling) | 0.525 |
| TimesNet | 0.685 |
| Autoformer | 0.300 |
| Non-stat. Transformer | 0.520 |
| FEDformer | 0.495 |
| Informer | 0.654 |
| *Multi-modal models* | |
| Toto+Qwen-3-4B | 0.826 |
| Toto+Qwen-3-4B+Thinking | 0.720 |

*Table 5.* Accuracy of the models in forecasting.

| Model | MAE | RMSE |
|---|---|---|
| *Foundation models* | | |
| Moment | 0.543 | 0.721 |
| Moirai2 | 0.516 | 0.698 |
| Toto | 0.489 | 0.675 |
| *Time series models* | | |
| Mantis | 0.457 | 0.603 |
| Mantis (w/o scaling) | 0.570 | 0.843 |
| TimesNet | 0.159 | 0.396 |
| Autoformer | 0.458 | 0.894 |
| Non-stat. Transformer | 0.256 | 0.561 |
| FEDformer | 0.171 | 0.408 |
| Informer | **0.143** | **0.358** |
| *Classical baselines* | | |
| DLinear | 0.696 | 1.073 |
| per-KPI Lin Reg | 0.162 | 0.395 |

vide a failure case shared across all evaluated models in Fig. 6. As shown, an increase in TX_Bytes, a typical pattern observed during streaming applications, triggers a false positive anomaly prediction, irrespective of the behavior of other channels. This highlights the difficulty these models face in handling naturally abrupt, yet normal, behaviors that are prevalent in practical observability scenarios.

Next, when additional context is provided, models are less likely to misclassify naturally fluctuating samples as anomalous. However, precision remains low, indicating that models still struggle to distinguish between normal erratic behavior and true anomalies. This challenge extends to time series foundation models: despite pretraining on large-scale time series datasets, their representations fail to capture this subtle distinction. Although they surpass the performance of LLMs, the overall performance remains suboptimal, highlighting the difficulty of real-world observability data.

Additionally, with respect to the time series models, our results show that most architectures struggle to achieve strong performance on the dataset. A notable exception is Mantis (Feofanov et al., 2025), which embeds scale information (specifically, the mean and standard deviation of each patch) into its representations. This design allows the model to remain aware of the absolute values of the time series rather than relying solely on normalized trends, as is the case

for other architectures. This is further confirmed by the performance drop observed when scale information is removed from Mantis by removing the NME scalar encoders found in their architecture (Lin et al., 2024). More broadly, these results highlight that many observability anomalies are inherently magnitude-driven and become ambiguous under normalization. By preserving raw scale information, `TelecomTS` enables evaluation of this critical failure mode, which is often obscured in existing observability datasets.

Finally, the Toto-based multi-modal model achieves performance close to the original Toto architecture, though slightly lower due to the increased optimization difficulty introduced by integrating the language model backbone. Its reasoning-enabled variant shows a small additional drop, since part of the model's capacity is devoted to intermediate reasoning. In return, it gains explicit reasoning capabilities over the time series signals.

### 4.2. Anomaly Duration Analysis

In this task, we go beyond simple anomaly detection and evaluate the models' ability to localize the duration of anomalies within a sample. To this end, we consider only anomalous samples and present them to the models. Language models are prompted to identify the specific time series segment where the anomaly occurs. For the time series models, anomaly predictions are generated per KPI (i.e., covariate) at each timestamp. These predictions are then aggregated via majority voting across variates for each observation to determine the anomalous segment. Outputs are compared against ground truth to compute precision, recall, and F1 score. Results are reported in Table 3.

**Results Analysis**. Models perform relatively well on this task, suggesting that when an anomalous sample is already provided, localization of the anomalous segment becomes

*Table 6.* Performance of the models on the question-answering task.

| Model | Time series QA | | | | | Network QA | | | |
|---|---|---|---|---|---|---|---|---|---|
| | Statistics | | Periodicity | | Trends | Traffic | Mobility | Location | Congestion |
| | $MAE_{min}$ | $MAE_{max}$ | $MAE_{min}$ | $MAE_{max}$ | Acc | Acc | Acc | Acc | Acc |
| GPT-4.1 | 0.163 | 1588.1 | 57.61 | 93.01 | 0.163 | 0.448 | 0.533 | 0.294 | 0.494 |
| Claude 3.7-Sonnet | 0.093 | 1315.8 | 32.04 | 64.04 | 0.109 | 0.414 | 0.950 | 0.428 | 0.461 |
| o4-mini | 0.027 | 247.1 | 37.21 | 63.15 | 0.134 | 0.433 | 0.767 | 0.367 | 0.494 |
| DeepSeek-R1 | 0.020 | 1542.6 | 50.33 | 61.73 | 0.134 | 0.357 | 0.983 | 0.339 | 0.483 |
| Toto+Qwen-3-4B | $3.343 \times 10^{-4}$ | 8569.1 | 5.85 | 40.60 | 0.670 | 0.988 | 0.992 | 0.400 | 0.936 |
| Toto+Qwen-3-4B+Thinking | $7.101 \times 10^{-4}$ | 9120.4 | 6.42 | 44.10 | 0.632 | 0.978 | 0.965 | 0.388 | 0.894 |

more accurate. All in all, this highlights the importance of addressing false positives in anomaly detection, as performance can be greatly improved once normal fluctuations are distinguished from true anomalies. This further highlights the current limitation of foundation models in practical observability settings and the potential room for improvement.

### 4.3. Root Cause Analysis

In this task, we evaluate each model's ability to identify the root cause of an anomaly by distinguishing between anomaly types. Both language and multi-modal models are given anomalous samples and asked to predict the anomaly type among the predefined classes. A caveat in the contextual setting is that the model is additionally provided with information about the KPIs typically affected by each anomaly. Results are summarized in Table 4.

**Results Analysis**. Language models perform poorly, struggling to accurately distinguish between different anomaly types, even with contextual information (although performance does improve with context). Time series models equipped with trained classification heads perform better. Notably, Toto performs especially well due to being pretrained on diverse observability data trends, which facilitates effective transfer learning for this task. The Toto-based multi-modal model performs on par with Toto, with gains capped by the absence of additional textual context and increased optimization complexity, while its reasoning-enabled variant maintains comparable accuracy while acquiring explicit reasoning capabilities.

### 4.4. Forecasting

Given that language models are known to perform poorly on forecasting tasks (Tan et al., 2024), we focus our analysis on time series models. The performance results of these models are presented in Table 5.

**Results Analysis.** As observed, model performance varies considerably, with some models outperforming others. How-

ever, these metrics fail to capture the inherent difficulty of forecasting observability data, which exhibits long stable periods punctuated by abrupt spikes (details can be found in Appendix H). While models accurately predict constant regimes, they consistently fail to capture peaks due to delayed detection, magnitude errors, or oscillatory instability. Consequently, MAE is often inflated by stable segments, obscuring the core forecasting challenge in such environments. These characteristics highlight the need for specialized modeling approaches in this setting.

### 4.5. Time Series and Network-Level Q&A

As a final task, we evaluate models on time series and network-level question answering by providing natural language questions with time series data. Using the Q&A samples outlined in Section 3.3, we design an evaluation pipeline that measures performance using either mean absolute error or accuracy, depending on the type of question. Since the time series component of the Q&A task is structured by covariates, and for ease of presentation, we report results using the KPI that achieves the best and worst performance (in terms of MAE) within the relevant task category. For the network-level Q&A tasks, each model is provided with contextual information regarding the locations zones and overall network configuration. The results of this evaluation are summarized in Table 6.

**Results Analysis**. The results from this task reveal two key insights. First, in the context of time series Q&A, the model performs well on KPIs that exhibit smooth and stable behavior, where abrupt changes are minimal. However, for more erratic KPIs, particularly TX_Bytes, which naturally exhibits abrupt behavior, the model struggles to make meaningful predictions. This highlights a significant gap in current foundation models ability to analyze statistical characteristics of complex observability signals. Second, with regard to the network-level Q&As, while some models show reasonable performance-especially reasoning ones, they still fall short in effectively linking engineering concepts and the provided contextual knowledge to the underlying time

*Table 7.* Effect of removing scale information on task performance. Each cell reports accuracy without scale $\rightarrow$ with scale; anomaly detection reports F1. Largest drop per task highlighted in **bold**.

| Model | Root Cause | Anom. Det. (F1) | Location | Traffic | Congestion | Mobility |
|---|---|---|---|---|---|---|
| TimesNet | $0.688 \rightarrow 0.756$ | $0.462 \rightarrow 0.740$ | $0.913 \rightarrow 0.923$ | $0.993 \rightarrow 0.997$ | $0.989 \rightarrow 0.991$ | $0.972 \rightarrow 0.992$ |
| Informer | $0.656 \rightarrow 0.800$ | $0.451 \rightarrow 0.852$ | $0.915 \rightarrow 0.929$ | $0.993 \rightarrow 0.996$ | $0.992 \rightarrow 0.986$ | $0.959 \rightarrow 0.994$ |
| FEDformer | $0.492 \rightarrow 0.620$ | $0.330 \rightarrow 0.713$ | $0.873 \rightarrow 0.926$ | $0.975 \rightarrow 0.988$ | $0.776 \rightarrow 0.845$ | $0.972 \rightarrow 0.972$ |
| Autoformer | $\mathbf{0.280 \rightarrow 0.584}$ | $\mathbf{0.318 \rightarrow 0.784}$ | $\mathbf{0.118 \rightarrow 0.424}$ | $\mathbf{0.578 \rightarrow 0.989}$ | $\mathbf{0.548 \rightarrow 0.824}$ | $\mathbf{0.622 \rightarrow 0.831}$ |
| NS Transformer | $0.516 \rightarrow 0.708$ | $0.483 \rightarrow 0.804$ | $0.891 \rightarrow 0.944$ | $0.974 \rightarrow 0.997$ | $0.995 \rightarrow 0.994$ | $0.950 \rightarrow 0.981$ |

series data. This highlights a critical gap in current models' ability to perform multi-modal reasoning, underscoring the need for models that can more effectively integrate temporal data with textual context. Finally, the Toto + Qwen-3-4B multi-modal model consistently outperforms language-only and reasoning models across most network-related Q&A tasks, achieving near-perfect accuracy in traffic and mobility classification as well as congestion inference, while remaining competitive on location Q&A. These results highlight the value of multi-modal training for observability tasks. At the same time, the smaller performance margins observed in location Q&A, as well as in certain time series Q&A tasks, underscore the importance of preserving absolute scale information. Notably, the Toto architecture does not explicitly encode such scale information; however, absolute signal magnitudes directly encode distance-related properties, which are critical for several downstream tasks, including location estimation. Yet, such key characteristics of KPIs become difficult to recover after normalization. This limitation contributes to the reduced performance margins in these tasks, and further reinforces the value of `TelecomTS` as it preserves absolute scale information and enables future scale-aware reasoning in observability settings.

### 4.6. Scale Ablation

A key characteristic of `TelecomTS` is preserving the absolute scale of KPI measurements. To quantify the impact of scale encoding, we conduct an ablation across five architectures. For each model, we compare two variants: a *base* that uses the encoder backbone as-is, and a *scale-aware* variant that adds a parallel NME branch, which takes the per-channel mean and standard deviation of the raw input, embeds them, and concatenates the resulting vector with the encoder's pooled feature before the classification head (Lin et al., 2024). Results are reported in Table 7.

Adding scale information consistently improves performance across nearly all models and tasks. The largest gains occur for Autoformer, which improves by up to 30.4 points on root cause identification and 41.1 points on traffic classification — reflecting that its decomposition-based architecture discards absolute magnitude entirely, making the NME branch critical. Informer and Non-stationary Transformer also show substantial gains on root cause and anomaly de-

tection. The consistent improvement in root cause identification across all architectures is particularly notable: distinguishing among eleven fault types with overlapping temporal signatures requires knowledge of absolute KPI operating ranges, as different faults manifest at characteristic magnitude levels that become indistinguishable without scale encoding. Therefore, these results confirm that absolute magnitude carries diagnostic information that temporal patterns alone cannot recover. These results validate a core motivation of `TelecomTS`: by preserving absolute scale, our dataset enables evaluation of a critical failure mode that normalized observability datasets obscure, and highlights the value of scale-aware architectures in practical observability settings.

## 5. Conclusion

This paper introduced `TelecomTS`, a large-scale, high-resolution, multi-modal dataset designed to bridge the gap between existing time series datasets and the complexities of observability systems. `TelecomTS` comprises millions of observations collected from a 5G communication network, incorporating categorical and heterogeneous covariates while capturing the erratic and bursty dynamics characteristic of observability environments. Evaluations of state-of-the-art models—including time series, language, and reasoning models—reveal consistent underperformance on `TelecomTS`. This underperformance stems primarily from their inability to handle the highly erratic patterns characteristic of observability data, as well as their lack of mechanisms to encode and leverage scale information—an aspect that is crucial in such scenarios. These findings underscore the pressing need for more robust and scale-aware time series foundation models capable of effectively operating in complex, real-world observability environments.

## Impact Statement

This paper presents work whose goal is to advance the field of machine learning. There are many potential societal consequences of our work, none of which we feel must be specifically highlighted here.

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

# A. Analysis of **TelecomTS** and Comparison With Existing Datasets

While there exists a plethora of time series datasets, most multivariate datasets lack one or more of the following characteristics crucial for observability data: heterogeneous variates, fine-grained resolution, and categorical variates. Here, we present some commonly used datasets and compare them to TelecomTS.

**Heterogeneous Variates.** We display two case studies that highlight the homogeneity of existing multivariate datasets. The ETTh$_1$ dataset contains data from electrical transformers aggregated on one-hour intervals (Zhou et al., 2021). We provide 6 of its 7 variates in Figure 7. As can be seen, the variates share common behavior. Particularly, all variates exhibit monotonic trends and have similar high-frequency dynamics, with spikiness and sharp turning points. For example, by observing the HUFL and LUFL variates, we see a similarity in the peaks and troughs of the variates, and this loosely holds across most variates as well. Moreover, semantically, these variates are also similar since six of them measure six different types of power load. For instance, HUFL measures High Useful Load, LULL measures Low Useless Load, and MUFL measures Middle Useful Load. Only OT, which measures Oil Temperature, is meaningfully different.

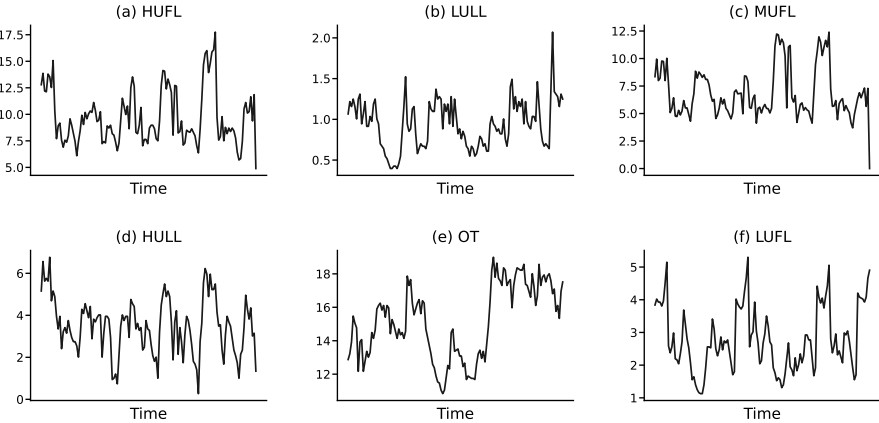

*Figure 7.* Randomly sampled variates from the ETTh$_1$ dataset.

Next, we observe the MotorImagery dataset that collects EEG data of imagined body movements using an $8 \times 8$ platinum electrode grid. Each of the 64 sensors corresponds to a variate, and data is recorded every millisecond. While the variates shown in Figure 8 display different trends, we see that they largely behave similarly. The variates lack volatility and exhibit minimal noise and no significant fluctuations or erratic behavior. Particularly, we can see low-frequency structure that spans significant portions of the interval, and we generally have smooth dynamics. Semantically, there is no diversity as all variates represent the same sensor measurement, just at different locations.

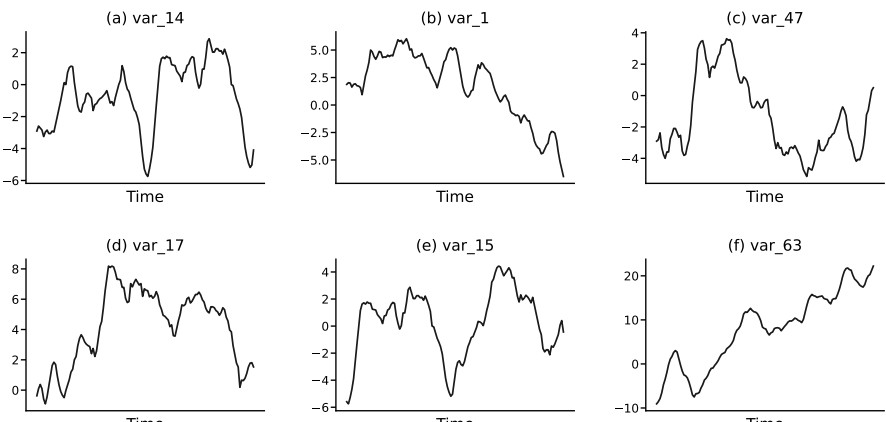

*Figure 8.* Randomly sampled variates from the MotorImagery dataset.

The above two examples were some of the many datasets that exhibited homogeneity across their variates. In fact, in our experiments, we selected multivariate datasets containing more than six variates from commonly used time series foundation model datasets, including the Unified Time Series Dataset, LOTSA, and others (Liu et al., 2024b; Woo et al., 2024). We then randomly sampled six variates and time series segments of 128 timestamps. We found that the vast majority of these

samples resembled the aforementioned examples, lacking sufficient diversity among their variates.

**Fine-Grained Resolution**. A large portion of existing time series datasets are temporally aggregated or averaged across multiple entities relevant to the scenario at hand. For example, climate metrics are typically aggregated at the monthly level, and energy usage data is frequently averaged across different cities. Such aggregation results in smoother and more predictable patterns, rendering these datasets unsuitable for observability applications that demand fine-grained resolution and the capacity to capture erratic, high-variance dynamics.

When it comes to temporal aggregation, the bulk of the datasets used by time series foundation models are recorded hourly, daily, weekly, and monthly. For example, at best, Chronos and TimesFM are trained on time series of 5-minute and 10-minute granularities, respectively (Ansari et al., 2024; Liu et al., 2024b). Moirai uses datasets on the second/multi-second granularity, but these comprise only $0.054\%$ of observations (Woo et al., 2024).

To highlight the impact of such resolution on the behavior of the time series, we report the FRED-MD dataset, a macroeconomic dataset comprising 107 variates spanning categories such as consumption, labor, income, interest rates, and other economic indicators (McCracken & Ng, 2016). Notably, the data is collected on a monthly frequency, which helps illuminate long-term macroeconomic trends. As seen in Figure 9, this leads to smooth trends, where several variables exhibit strong, positive, and smooth trends, while others display low-frequency fluctuations with minimal abrupt changes. This is far from the erratic and high-variance environments encountered in observability applications.

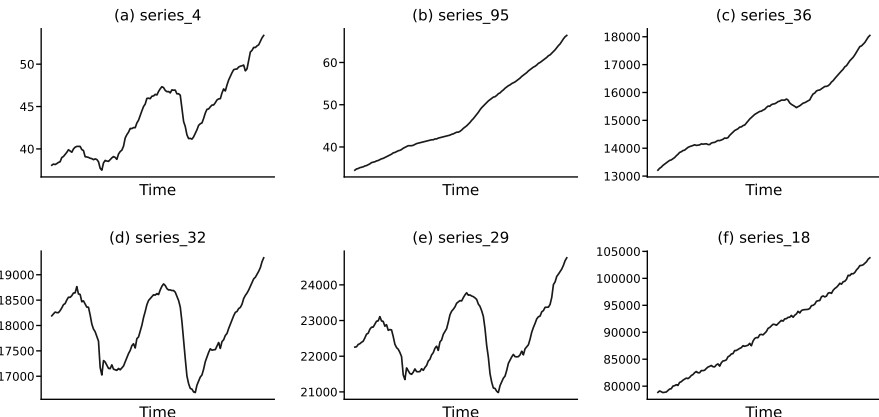

*Figure 9.* Randomly sampled variates from the FRED-MD dataset.

In another example, we report the Weather dataset that contains hourly data on temperature, humidity, wind, and other climate metrics (Zhou et al., 2021). Under the hourly frequency, we can see that daily or weekly trends dominate in Figure 10. In particular, DewPointFarenheit and DryBulbCelsius exhibit strong daily fluctuations, which can be too predictable and less relevant to erratic observability dynamics. The other variates also exhibit relatively smooth trends and low variance between consecutive timestamps.

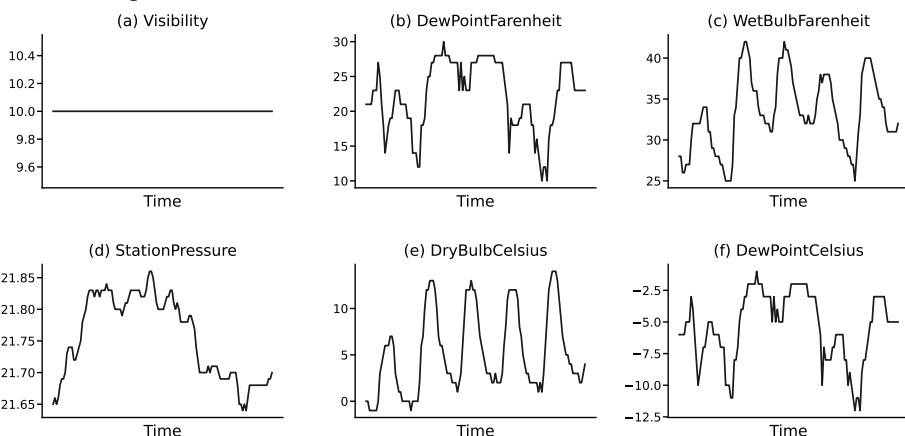

*Figure 10.* Randomly sampled variates from the WTH dataset.

Regarding spatial aggregation, many time series datasets collect data on the city, state, or even country level, which can

smooth out less predictable, high-frequency behavior. For example, the COVID Deaths dataset documents daily deaths from the COVID-19 pandemic where each time series corresponds to a whole country (Dong et al., 2020). Similarly, the CDC Fluview ILINet captures illness data on the state, regional, and national level (Centers for Disease Control and Prevention, 2017). Although spatial aggregation and temporal aggregation are often unsuitable for many observability applications, we note that there exists increasing interest in time series datasets with fine-grained spatial resolution due to the increased popularity of spatiotemporal data and distributed sensor deployment (Jiang et al., 2024).

**Multi-modal Downstream Tasks**. The lack of multi-modal time series datasets remains a significant bottleneck in the development of capable multi-modal time series foundation models. There exists few natively multi-modal datasets, most notably Time-MMD, while most datasets retroactively annotate existing time series data, such as in TIME-MQA which annotates datasets from UTSD (Liu et al., 2025; Kong et al., 2025b). As a result, many studies in this domain are forced to bootstrap their own datasets. Moreover, the majority of existing datasets are designed solely for forecasting tasks, with limited support for other practical applications such as anomaly detection or root cause analysis.

**Our Dataset.** From Figures 11 and 12, we see that our dataset starkly differs from the previously displayed examples. Firstly, UL_Protocol and DL_Protocol are both categorical variates that exhibit unique temporal and statistical dynamics. Following this, we see that we have high variate diversity. In our numerical data, we have low-frequency variates such as UL_MCS, DL_MCS, and RSRP, which may be less erratic. On the other hand, we have high variance and noisy variates with UL_SNR, UL_NPRB, etc. Even within our high variance variates, we have lots of diversity. We can see that UL_SNR has many sharp turns, frequent spikes and troughs. On the other hand, UL_NPRB is very spiky in one direction and often resets to a baseline value. Moreover, both RX_Bytes and TX_Bytes exhibit sporadic spikes at lower frequencies, typically corresponding to specific events—such as bursts in downloaded data. These observations exemplify the importance of fine-grained data, as such unpredictable spikes are not averaged or aggregated out at our 100ms time-scale. Finally, beyond the behavior of the covariates, it is important to note that our variates span multiple data types—ranging from integers (e.g., TX_Bytes) to floating-point values such UL_BLER—covering distinct range of values.

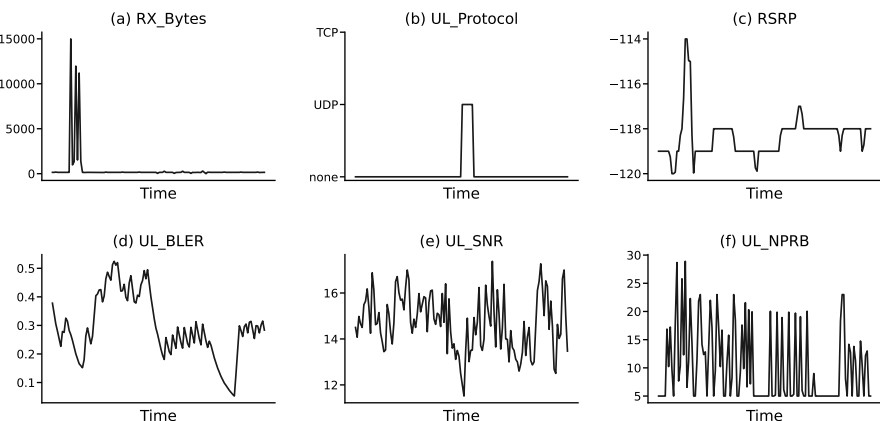

*Figure 11.* Randomly sampled sequence from `TelecomTS` (Example 1).

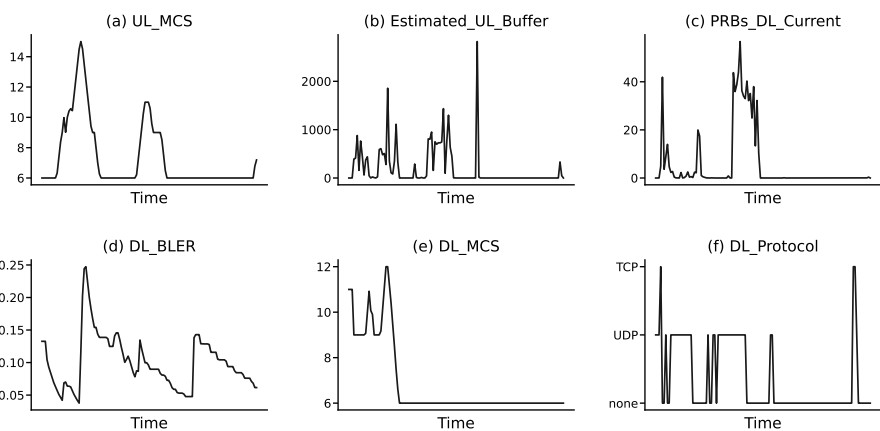

*Figure 12.* Randomly sampled sequence from `TelecomTS` (Example 2).

# B. Data Collection Details

## B.1. 5G Network

**Overview**. To facilitate the collection of fine-grained temporal and cross-layer network KPIs from a fully operational 5G system, we implemented a standalone 5G network deployed in a controlled lab environment capable of supporting real over-the-air transmissions and enabling diverse, repeatable experimental configurations. The network consists of a single monolithic base station, connected to a software-defined radio (SDR) with a radio unit (RU) for low-level physical layer processing and signal transmission. The SDR interfaces with the gNB via a dedicated 10 Gbps Ethernet fronthaul link. Additionally, the gNB connects to the 5G core network instance over standard N2/N3 interfaces through a separate 10 Gbps Ethernet backhaul link, enabling full end-to-end standalone operation. A visual overview of the network deployment is provided in Fig. 2, where (a) shows the mobile devices and the RU, and (b) illustrates the server-side infrastructure hosting the core network and baseband functions. While the system supports multi-band operation, all experiments in this work were conducted in the n78 TDD band, using a 38.16 MHz channel bandwidth centered at 3.319 GHz.

The gNB and core network were implemented using the latest release of the open-source OpenAirInterface (OAI) software stack (OpenAirInterface Software Alliance, 2024). The baseband processing stack of the gNB, including the PHY, MAC, RLC, PDCP, and RRC layers, was deployed on a high-performance server equipped with an AMD Ryzen Threadripper PRO CPU (4.4 GHz, 24 cores) and 128 GB of RAM. The 5G core network included all standard functional entities defined by 3GPP, including the access and mobility management function (AMF), session management function (SMF), user plane function (UPF), authentication server function (AUSF), network repository function (NRF), unified data repository (UDR), and unified data management (UDM). These components were deployed as containerized services within a high-availability Kubernetes cluster, hosted on a separate high-performance server with the same hardware specifications as the gNB host.

The RU was realized using a USRP N300 SDR, configured with two UBX daughterboards, each supporting up to 100MHz of instantaneous bandwidth per channel. To enhance directional transmission and reception, we utilized a 2×2 beamforming configuration provided by OAI. Finally, Google Pixel 6 and 7 smartphones, provisioned with programmable 5G SIM cards, served as the User Equipment (UE) throughout all experiments.

**Adversarial Environment Setup**. To further enhance the experimental environment and enable data collection under adversarial conditions, a suite of malicious jammers was implemented and integrated into the network. As shown in Fig. 2(c), a USRP X310 SDR was utilized to synthesize controlled over-the-air jamming signals using GNU Radio software, with the jammer strategically positioned at varying locations relative to the RU to emulate diverse radio link impairments. To introduce flexibility and realism into the adversarial environment, multiple jamming configurations were implemented, allowing dynamic control over transmission gain, occupied bandwidth, and jammer activity patterns. Throughout the campaign, three types of jamming attacks were generated: single-tone jamming (continuous narrowband interference at a specific frequency), pulsed jamming (intermittent bursts of narrowband interference), and wideband noise jamming (broad-spectrum interference across a wide frequency range). These jamming signals were transmitted over the air with the objective of disrupting the RU–UE communication link during the data collection process and observing the resulting impact across multiple KPIs, including signal quality, throughput, and error rates.

**Network Performance Tuning and Optimization**. To support long-duration experimentation and ensure reliable KPI collection with real-time granularity, several low-level software and hardware optimizations were required to maintain stable end-to-end network performance. During early operation, we observed recurring instability during measurements, often resulting in intermittent UE disconnections and incomplete KPI traces. This instability was primarily attributed to three factors: (i) the limited transmit power of the RU, which reduced link robustness during sustained over-the-air operation; (ii) processing bottlenecks on the gNB baseband stack, where high-rate IQ samples were occasionally delayed or dropped; and (iii) external in-band interference, which intermittently affected reception quality in the n78 band.

To address these challenges, we introduced a set of system-level optimizations targeting both the server running the baseband processing functions and the SDR. On the baseband server, we disabled hyper-threading to eliminate core contention, deployed a low-latency Linux kernel to reduce scheduling delays, disabled kernel page table isolation to mitigate Spectre-related overhead, and set the CPU governor to performance mode to maintain maximum CPU performance by preventing frequency scaling and disabling energy-saving states. On the RU side, the fronthaul link was carefully tuned to ensure deterministic and lossless IQ sample delivery. Jumbo frames with a Maximum Transmission Unit (MTU) of 9000 bytes were enabled to reduce packetization overhead, and both kernel socket buffers and Ethernet ring buffers were enlarged to accommodate high-throughput traffic without introducing jitter or packet drops. Finally, to minimize the impact of external

interference, the operating frequency within the n78 band was selected based on in-band noise measurements, allowing us to identify and utilize the cleanest available sub-band for over-the-air transmission.

### B.2. Data Collection and Preparation

**Network Zoning for Controlled Experiments.** To systematically capture KPI variations under diverse radio conditions, the network was deployed in a controlled lab environment covering approximately 70 m$^2$. The space was partitioned into three spatial zones—Zone A, Zone B, and Zone C—based on the distance between the UE and the RU. This zoning strategy enabled controlled experimentation across distinct wireless conditions, facilitating structured data collection for downstream analysis. Zones closer to the RU correspond to stronger signal conditions with minimal interference, while more distant zones experience weaker signals due to increased distance and potential obstacles.

**Zone A** includes all locations within a 3-meter radius of the RU, representing scenarios with strong signal strength, low path loss, and minimal fading.

**Zone B** spans distances between 3 and 6 meters, emulating moderate signal quality with potential variations due to partial obstruction or environmental reflections.

**Zone C** comprises all areas beyond 6 meters, corresponding to weak-signal conditions with increased attenuation, and a higher likelihood of radio link degradation.

A visual layout of the lab environment and spatial zoning configuration is shown in Fig. 13, illustrating the relative position of the RU and the boundaries of each zone.

**Application-Level Traffic and Interference Scenarios.** To capture network behavior under representative real-world conditions, we selected a suite of application-layer scenarios encompassing both typical user behavior and adverse operational contexts. Experiments were conducted under two UE mobility profiles: (i) a static profile, where the UE remained stationary, and (ii) a mobile profile, where the UE moved at a constant pedestrian speed of 5 km/h to emulate realistic urban mobility in the lab.

*Figure 13.* Spatial partitioning of the environment into 3 zones.

The selected applications reflect common mobile usage patterns while imposing varied demands on different layers of the network protocol stack. Specifically, in the mobile device we run a set of typical mobile applications during data collection, including buffered video streaming via YouTube, live video streaming via Twitch, and large file downloads over HTTP. In addition, to examine system performance under resource contention, we introduced a controlled congestion scenario by connecting a second UE executing concurrent download tasks, thereby increasing cell load during the data collection phase.

A detailed breakdown of the data collection observations for each traffic type, mobility pattern, and zone is presented in Table 8, capturing the spatiotemporal scope of the experimental campaign across all operating conditions.

To study network behavior under adversarial conditions, we conducted controlled data collection sessions with active jamming during live application traffic. In each session, the UE maintained continuous traffic flows while exposed to over-the-air interference from a co-located jammer, allowing us to observe both control- and data-plane KPIs under degraded radio conditions. The jammer remained stationary throughout the experiments, with its placement illustrated in Fig. 13. We employed multiple jamming patterns, including wideband noise covering the full n78 band, single-tone, and pulse-based interference. The jamming signal followed a periodic pattern, alternating between 2 seconds of activity and 8 seconds of silence. To ensure effective disruption of the RU–UE link, the jammer's transmit gain was set to 25 dBi. Representative spectrograms showcasing benign and jammed scenarios are presented in Fig. 14. These include samples of wideband noise and pulsed interference patterns used during the experiments.

**Data Collection, Filtering, and Synchronization**. For each measurement scenario, a single mobile device was connected to the network and actively engaged in the designated traffic session for a continuous duration of four hours. During each session, data was collected at both the link and the network layer to enable detailed analysis of network behavior. To isolate relevant traffic, all control plane signaling was excluded from the dataset. In the user plane, only transport-layer headers (i.e., TCP and UDP) were retained, while payload data was discarded to reduce storage overhead. Packet-level information was captured using Wireshark on the core network, enabling inspection of traffic characteristics and flow-level behavior, followed by IP filtering to isolate the target device.

| Category | Condition | | Activity | Zones | Observations |
|---|---|---|---|---|---|
| Normal | Static | No congestion | YouTube | A, B, C | 100k/zone |
| | | | Twitch | A, B, C | 100k/zone |
| | | | File | A, B, C | 100k/zone |
| | | Congestion | YouTube | A, B, C | 10k/zone |
| | | | Twitch | A, B, C | 10k/zone |
| | | | File | A, B, C | 10k/zone |
| | Motion | | YouTube | n/a | 10k |
| | | | Twitch | n/a | 10k |
| | | | File | n/a | 10k |
| Anomalous | Jamming | | YouTube | A | 10k |
| | | | Twitch | A | 10k |
| | | | File | A | 10k |
| | Synthetic | | YouTube | A, B, C | 10k/zone |
| | | | Twitch | A, B, C | 10k/zone |
| | | | File | A, B, C | 10k/zone |

*Table 8.* Breakdown of total number of observations across all zones and experimental conditions.

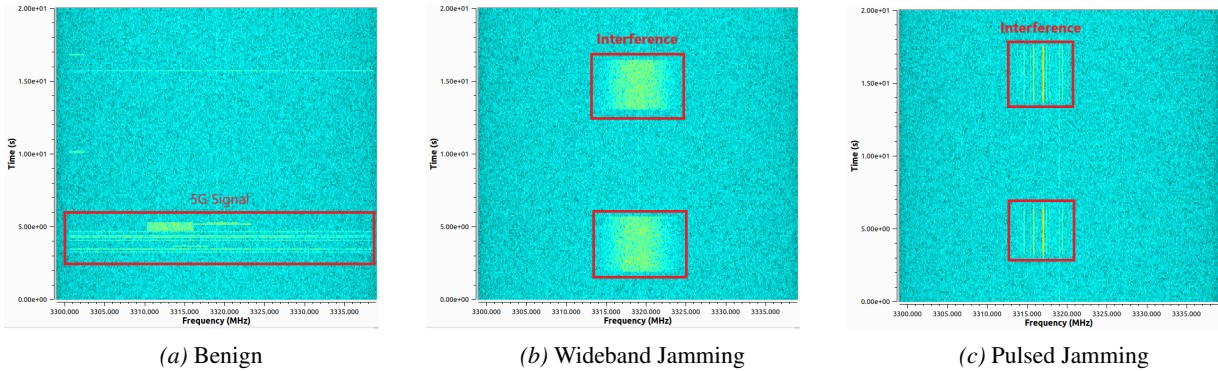

*(a)* Benign                *(b)* Wideband Jamming                *(c)* Pulsed Jamming

*Figure 14.* Spectrograms illustrating benign and adversarial interference patterns during collection.

Due to independent timestamping mechanisms between the link layer logging modules and the packet capture software, a temporal misalignment existed across the two data sources of the order of two seconds. To address this, we found the time offset that best matched the transmitted byte counts (from the Physical layer) with the downlink packet counts (from the network trace), which are expected to be highly correlated, and applied it to all PHY-layer KPIs to synchronize them with the network-layer trace. For this, we used a histogram-based matching technique: for each 300 ms window of the transmitted byte series, we computed the KL divergence against 300 ms windows of the packet count series, slid with a 30 ms stride. The best-matching offset for each window was recorded, and the mode of these offsets was selected as the final alignment correction.

To ensure consistency and repeatability across experiments, we followed a structured procedure that orchestrated each stage of the data collection pipeline—from system initialization to postprocessing. The detailed steps of this end-to-end process are outlined in Algorithm 1, which captures the sequence of operations for initializing the 5G network, configuring devices, capturing KPIs and traffic, and exporting the synchronized dataset for analysis.

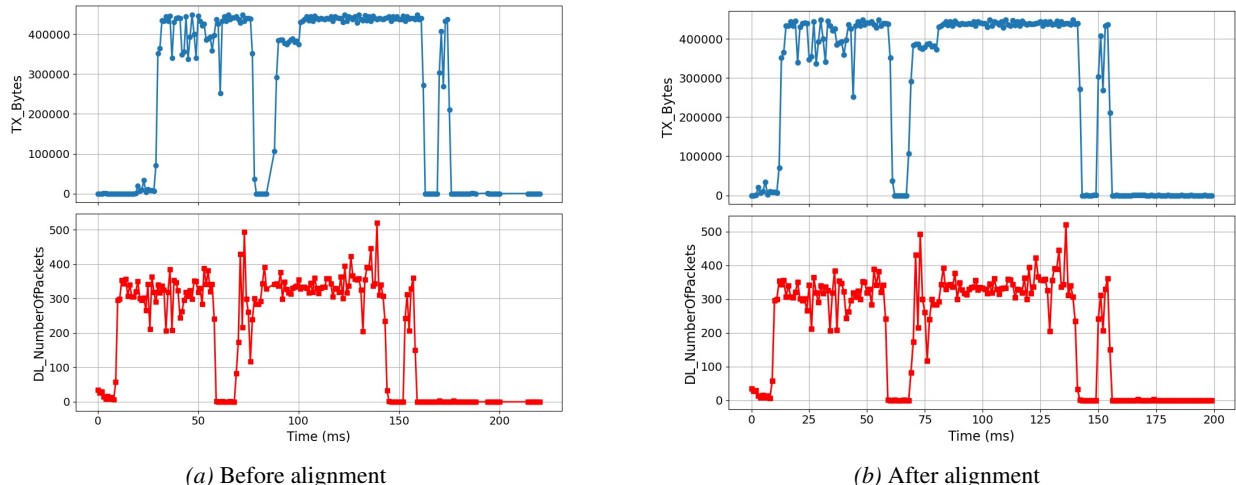

*(a)* Before alignment                                  *(b)* After alignment

*Figure 15.* Number of packets (top) and number of transmitted bytes (bottom) before (a) and after (b) alignment.

---

**Algorithm 1** End-to-End Data Collection Procedure

---

 1: **Initialize Core Network**
 2:     Deploy containerized 5G Core components as pods in the Kubernetes Cluster
 3:     Verify inter-component connectivity between the pods
 4: **Configure Radio Unit**
 5:     Set sampling rate, transmit gain, and center frequency
 6:     Ensure proper synchronization and signal lock
 7: **Activate gNB**
 8:     Launch baseband processing stack
 9:     Establish registration and connection to the Core network
10: **Start Data Logging Modules**
11:     Activate KPI loggers on gNB and UE
12:     Start user plane packet capture on the Core network
13: **Connect Mobile Device**
14:     Power on mobile device and disable airplane mode
15:     Attach device to the network and establish user session
16:     Verify IP configuration and data-plane reachability
17: **Run Traffic Session**
18:     Generate traffic via selected application (YouTube, Twitch, File Downloading)
19:     Maintain session for the experiment duration
20: **Postprocessing**
21:     Filter control-plane traffic, discard payload data, and retain only packets associated with the target device IP
22:     Parse KPIs and packet logs
23:     Apply timestamp alignment (e.g., histogram-based matching)
24:     Export synchronized dataset for downstream analysis

---

**Overview of Collected KPIs**. To enable fine-grained monitoring of wireless performance across all protocol layers, our dataset includes a rich set of KPIs captured from both the base station and the mobile device. These KPIs span the physical (PHY), medium access control (MAC), and network layers, providing a multi-dimensional view of network behavior under varying radio, mobility, and interference conditions. The metrics include signal quality indicators, resource allocation statistics, error rates, transport protocol usage, and traffic volume, allowing detailed analysis of both control and data plane dynamics. The tables below summarize each collected KPI, including a brief description of each for reference.

**RSRP (Reference Signal Received Power)**

**Layer:** PHY      **Reported by:** UE      **Type:** Numerical (float)      **Range:** [-140, -45]
Measures the received signal strength from the base station's reference signals. Reflects path loss and coverage quality.

**UL_SNR (Uplink Signal-to-Noise Ratio)**

**Layer:** PHY      **Reported by:** UE      **Type:** Numerical (float)      **Range:** [-3.5, 60]
Indicates the uplink signal quality at the receiver. A higher SNR corresponds to better link reliability.

**DL_BLER / UL_BLER (Downlink / Uplink - Block Error Rate)**

**Layer:** MAC      **Reported by:** gNB      **Type:** Numerical (float)      **Range**: [0, 1]
Fraction of erroneous transport blocks over total transmitted blocks in downlink/uplink. High BLER signals poor radio conditions.

**DL_MCS / UL_MCS (Downlink / Uplink - Modulation and Coding Scheme)**

**Layer:** MAC      **Reported by:** gNB      **Type:** Numerical (float)      **Range**: [0, 27]
Represents the average modulation and coding level selected for a given link. Higher values indicate more aggressive transmission schemes.

**UL_NPRB (Allocated Uplink Physical Resource Blocks)**

**Layer:** MAC      **Reported by:** gNB      **Type:** Numerical (int)      **Range**: [0, 105]
Number of Physical Resource Blocks assigned to the UE for uplink transmission during a Transmission Time Interval.

**Estimated_UL_Buffer**

**Layer:** MAC      **Reported by:** gNB      **Type:** Numerical (int)      **Range**: [0, 250k]
Estimation of buffered uplink data at the UE as reported to the gNB via Buffer Status Reports.

**PRBs_DL_Current / PRBs_UL_Current (Downlink / Uplink - Physical Resource Blocks)**

**Layer:** MAC      **Reported by:** gNB      **Type:** Numerical (float)      **Range**: [0, 105]
Number of Physical Resource Blocks currently allocated to the UE in the downlink/uplink direction in a given Transmission Time Interval.

**PRB_Utilization_DL / PRB_Utilization_UL (Downlink / Uplink - Physical Resource Block Utilization Ratio)**

**Layer:** MAC      **Reported by:** gNB      **Type:** Numerical (float)      **Range**: [0, 100]
Percentage of total Physical Resource Blocks utilized by the UE in downlink/uplink over time, indicating traffic load and resource usage.

**TX_Bytes / RX_Bytes (Transmitted / Received Bytes)**

**Layer:** MAC      **Reported by:** gNB      **Type:** Numerical (int)      **Range**: [0, 450M]
Total number of user-plane bytes transmitted and received, used to compute throughput and volume.

> **UL_Protocol / DL_Protocol (Uplink / Downlink - Transport Protocol)**
>
> **Layer:** Network     **Reported by:** UPF     **Type:** Categorical     **Range**: {TCP, UDP, None}
> Specifies the transport protocol (TCP or UDP) used in the uplink/downlink direction.

> **UL_NumberOfPackets / DL_NumberOfPackets (Uplink / Downlink - Packet Count)**
>
> **Layer:** Network     **Reported by:** UPF     **Type:** Numerical (int)     **Range:** [0, 10k]
> Total number of user-plane packets observed in the uplink/downlink direction.

## C. Anomaly Curation Details

**Modeling KPI Effects**. To simulate an anomaly, we apply transformations to our collected wireless data. Specifically, we alternate between sampling from two exponential distributions, one to get the anomaly inter-arrival time (the time between two anomalies) and one to get the anomaly duration. We adopt exponential priors following established real-world behavior in wireless and networked systems (Maatouk et al., 2024), where anomaly inter-arrival times and durations have been repeatedly observed to exhibit memoryless characteristics and weak temporal dependence. Importantly, our results are not sensitive to this specific choice: since anomalies are segmented into fixed-length samples, changing the prior solely affects the within-sample duration distribution, while downstream task performance on anomaly detection, root cause analysis, and other tasks remains similar. This gives us a set of timestamps $(s_i, t_i)$ of anomaly start and end time pairs. For each timestamp $(s_i, t_i)$, we will assign it some anomaly type $a_i$. Then, for the relevant affect KPIs affected by this anomaly type $(v_j^{s_i}, \ldots, v_j^{t_i})$, we apply a function to get our transformed data, $(f(v_j^{s_i}), \ldots, f(v_j^{t_i}))$.

As anomalies can have diverse effects on KPIs, as evidenced by the gathered scholarly material, we use 8 different function types listed in Table 9. Two of such function types, constant addition and multiplication, are static, while the other functions evolve with time. Next, given that many KPIs have a fixed range of possible values (e.g., UL_BLER can only be between 0 and 1), we assign boundaries to all KPIs when appropriate and truncate any values that exceed these boundaries. This often occurs with transformations such as exponential growth, where KPIs experience saturation, usually signaling a severe anomaly.

*Table 9.* List of Function Types

| Function Type | Temporal | Description | Parameters |
| --- | --- | --- | --- |
| Constant Addition | ✗ | Add a fixed constant to all points | Additive Shift |
| Constant Multiplication | ✗ | Multiply all points by a fixed factor | Multiplicative Factor |
| Linear Growth | ✓ | Increase linearly | Slope |
| Exponential Growth | ✓ | Multiply data by an exponential | Growth Rate |
| Logistic Growth | ✓ | Add a logistic growth function | Growth Rate |
| Logarithmic Decay | ✓ | Multiply by decay factor | Decay Rate |
| Sinusoidal Fluctuation (additive) | ✓ | Add a sine function | Amplitude, Frequency, Shift |
| Sinusoidal Fluctuation (multiplicative) | ✓ | Multiply by a sine function | Amplitude, Frequency, Factor |

However, truncating to these "hard" boundaries can often be unrealistic, as most systems will fail before reaching theoretical saturation and not all anomalies manifest as saturations. Therefore, we set a range of soft bounds for each KPI and sample from these ranges to get our threshold. To avoid excessively aggressive soft bounds that truncate non-anomalous data, we take the threshold to be the minimum or maximum with respect to the 20th lowest or largest data point in the input time series. We choose the number 20 empirically to avoid outliers or measurement errors that may result in an issue like UL_BLER being equal to 1.06. Furthermore, when a KPI reaches saturation, we inject noise at the saturated data points, as otherwise, we get unrealistic flat clipped values.

For specific function classes, such as sinusoidal fluctuations, linear growth, and logistic growth, we may choose to inject small noise to maintain realism for these additive effects. Additionally, a naive implementation of exponential growth leads to incredibly noisy data, often due to the presence of 0 or other small values in the data, as the data will jump between exponentially high values and near 0 within a few timestamps. For example, the transmitted bytes may be very high in

general. However, for a given decisecond, it is possible that no bytes are transmitted. We remedy this by first injecting small positive noise before multiplying by the exponential factor. Furthermore, small variations in the natural noise of our data will blow up under exponential growth. Therefore, we apply kernel smoothing to get the general trend of our KPI and subtract the kernel-smoothed values from our original values to get residuals. Then, we apply exponential growth to our smoothed values and add back our residuals. This leads to realistic exponential growth behaviors with appropriate variance.

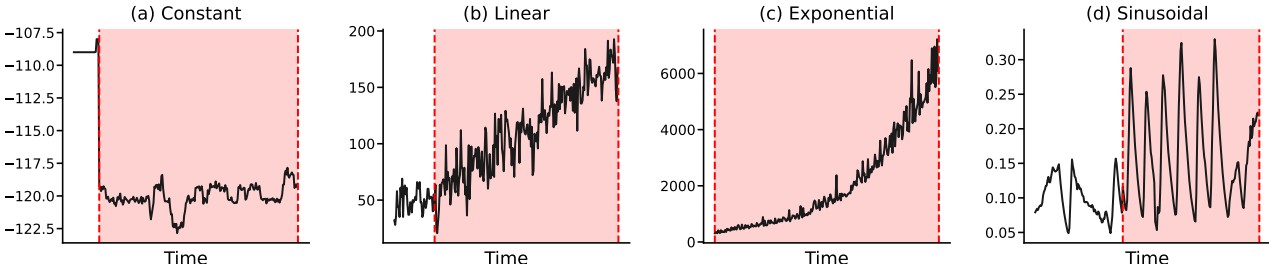

*Figure 16.* Examples of anomaly effects under varying function types.

**Anomaly Curation**. We carefully select a list of 10 representative anomalies listed in Table 10 that span diverse effects on KPIs. These anomalies can be classified into one of five types of wireless anomalies: hardware failure, software issues, infrastructure issues, environmental interference, and anomalous usage. Three of our anomalies are static, meaning that all KPIs are affected statically, reflecting sudden onset anomalies. The remaining anomalies are temporal and represent anomalies that gradually build up. For every anomaly, we use scholarly material to select a list of KPIs that would be affected under this anomaly and match each KPI with a function class/KPI effect mentioned in the previous section. Most importantly, to accurately simulate anomalies, we carefully pick parameters for these functions classes to match the anomaly. Given an affected KPI of an anomaly, we use GPT-4.1 to generate a range of feasible values for each parameter. Using a range allows us to model the stochasticity of real-world anomalies. Then, at generation time, we will uniformly sample within this range to determine the transformation applied to our data. We repeatedly verify these parameters using human feedback until we have satisfactory results.

*Table 10.* List of Anomalies

| Anomaly | Domain | Temporal | Affected KPIs |
|---|---|---|---|
| Antenna Failure | Hardware | ✓ | 13 |
| Buffer Overflow | Software/Infrastructure | ✓ | 10 |
| Co-Channel Interference (Mild) | Infrastructure | ✗ | 9 |
| Co-Channel Interference (Severe) | Infrastructure | ✗ | 11 |
| Doppler Shift | Environment | ✓ | 7 |
| Faulty Handover Algorithm (Frequent) | Software | ✓ | 9 |
| Faulty RF Filters | Hardware | ✓ | 9 |
| High Network Congestion (Static) | Usage | ✗ | 10 |
| High Network Congestion (Temporal) | Usage | ✓ | 10 |
| Resource Allocation Bugs | Software | ✓ | 9 |

**Troubleshooting Ticket**. We provide GPT-4.1 with the prompt found in Appendix D to generate a troubleshooting ticket each time we simulate an anomaly in our wireless data. The anomaly impact variable inputs a fixed textual description for each anomaly that lists how every affected metric changes under the anomaly. The alarm time and resolution time are optional inputs that match the start and end time of the corresponding anomaly. We use a human-in-the-loop process to ensure quality for our tickets.

**Anomaly Dataset.** To obtain a final anomaly dataset, we slice all anomalous time series sequences into sequences of length 128 with a stride of 32. We remove all samples that no longer contain anomalous data points. We also remove all

samples containing two different types of anomalies to simplify downstream tasks and to avoid unrealistic anomaly density, as recommended by (Wu & Keogh, 2023). Finally, we add relevant metadata such as alarm and resolution time, affected metrics, and an indicator array for anomalous data points.

## D. Prompts

**Troubleshooting Ticket.** This prompt is used to generate multi-modal anomaly detection and root cause analysis simultaneously with the simulated time series anomaly data in Appendix C. For every instance of an anomaly, we record its type (for instance "Antenna Failure") as well as its impact on metrics. The latter is a paragraph that describes how each affected KPI is transformed under the anomaly and its corresponding function type. Optionally, we can include the alarm and resolution time. We prompt GPT-4.1 to perform a root cause analysis and generate a hypothetical solution to the simulated anomaly. Such troubleshooting tickets can hopefully endow multi-modal anomaly detection models with anomaly analysis and resolution insights.

---

**Troubleshooting Ticket Generation Prompt**

Generate a concise troubleshooting summary for a wireless network anomaly.
**Context:**

- **Anomaly Type:** `anomaly_type`

- **Alarm Time:** `anomaly_time`   **[Optional]**

- **Resolution Time:** `resolved_time`   **[Optional]**

- **Anomaly Impact:** `anomaly_impact`

**Format the response as follows (DO NOT add extra explanations):**
**Diagnose Summary:**

- **Issue:** [Briefly describe the detected anomaly.]

- **Symptoms:** [Summarize affected metrics and key changes.]

- **Root Cause:** [State the most likely cause.]

- **Resolution:** [Summarize the main actions taken to fix it.]

---

**Anomaly Experiment Prompts.** The following prompts are used to assess a model's anomaly detection and analysis capabilities. For each prompt, we provide a time series sample as well as the alarm and resolution times. Detailed instructions and context are given, and we optionally provide additional context on wireless data or anomaly descriptions to aid the model. Finally, we prompt the model to output a strictly formatted conclusion block that allows for regular expression parsing.

---

**Anomaly Detection Prompt**

You are an AI assistant tasked with analyzing time series data for anomalies in a wireless network. You will be provided with a time series dataset containing various metrics and a specific time range to analyze. The time series is sampled every 0.1 seconds (i.e. timestamps are a decisecond apart), and contains a total of $n$ time steps. Your goal is to detect any anomalies within this range and identify the timestamps where they occur.

**[Optional, if `context=True`]**

Note: Wireless network data is naturally **noisy and erratic**, even under normal conditions. Sporadic spikes, sharp drops, or momentary fluctuations can appear **without indicating any true anomaly**. This sequence is only $\ell$ seconds long, so be especially cautious in interpreting short-term changes as significant. Only mark something as anomalous if there is **clear and sustained evidence** of abnormal behavior across multiple metrics.

First, review the time series data provided from `start_time` to `end_time`:

```
metric_1: v_1^1 v_2^1 ... v_n^1
metric_2: v_1^2 v_2^2 ... v_n^2
more metrics ...
```

**[Optional, if `context=True`]**

To detect anomalies, follow these steps:

1. Begin by scanning the time series for any unusual behavior: sharp spikes or drops, sustained deviations, or values inconsistent with the expected range.

2. Consider inter-metric relationships — for example, whether high buffer utilization coincides with low throughput or high BLER.

3. All anomalies occur at the same timestamp range, so you should identify a single set of timestamps for the anomaly event and attribute affected metrics to that period.

Summarize your conclusion as follows:

```
<conclusion>
Anomaly Detected:  [Yes/No]
[If yes, include the following strictly formatted line:]
Anomaly Timestamps:  [(start_time1, end_time1), (start_time2,
end_time2), ...]
</conclusion>
```

Only base your analysis on the provided time range. If no anomaly is detected, write:

```
<conclusion>
Anomaly Detected:  No
</conclusion>
```

Do not include additional comments or summaries outside this format.

---

**Anomaly Boundary Prompt**

---

You are an AI assistant tasked with analyzing time series data for anomalies in a wireless network. You will be provided with a time series dataset containing various metrics and a specific time range to analyze. The time series is sampled every 0.1 seconds (i.e. timestamps are a decisecond apart), and contains a total of $n$ time steps. Your goal is to identify a **single contiguous time interval** during which an anomaly occurs. There is exactly one anomaly in the data, and it may span the entire sequence or just a sub-segment.

First, review the time series data provided from `start_time` to `end_time`:

    metric_1: $v_1^1$ $v_2^1$ ... $v_n^1$
    metric_2: $v_1^2$ $v_2^2$ ... $v_n^2$
    more metrics ...

Summarize your conclusion as follows:

    <conclusion>
    Anomaly Timestamps:  (YYYY-MM-DD HH:MM:SS.sss, YYYY-MM-DD
    HH:MM:SS.sss)
    </conclusion>

Do not include any additional commentary or explanation outside the specified format. Respond with *only* the `<conclusion>` block and nothing else.

---

**Root Cause Analysis Prompt**

---

You are an AI assistant tasked with diagnosing a known anomaly in wireless network time series data. You will be provided with a short time series segment sampled every 0.1 seconds, covering $\ell$ seconds and $n$ time steps. This sequence ranges from `start_time` to `end_time` and **is confirmed to contain an anomaly**.

The anomaly is known to be **one of the following**, and each is equally likely to occur in this dataset. **Do not assume any anomaly is more common or more likely than another.**

Your task is to identify the most plausible anomaly type **from the following list:** `anomaly_list`

Please analyze the metrics below and select the **single most likely anomaly**.

**[Optional, if `descriptions=True`]**
Here is a summary on how the provided anomalies generally behave: `[Anomaly descriptions]`

Here is the time series data:

    metric_1: $v_1^1$ $v_2^1$ ... $v_n^1$
    metric_2: $v_1^2$ $v_2^2$ ... $v_n^2$
    more metrics ...

Summarize your conclusions as follows:

    <conclusion>
    Anomaly Type:  [One exact string from the predefined anomaly
    list.]
    </conclusion>

Do not include any additional commentary or explanation outside the specified format. Respond with *only* the `<conclusion>` block and nothing else.

---

**Time Series QA Prompts**. These prompts are used to assess a model's time series analysis capabilities. For each prompt, we provide a single KPI from a sample and ask the model to perform elementary statistical analysis such as detecting the average value, variance, etc. Oftentimes, models will output reasoning steps, so we include warnings to discourage such behavior, which has significantly helped with regular expression parsing.

---

**Mean Detection Prompt**

---

Consider the following list of numbers representing a time series: $v_1, v_2, \ldots, v_n$. Some values may be missing (NaN). What is the average `channel` value of this series, ignoring NaNs? Respond with only a single float rounded to 2 decimal places — no other text or numbers. Please DO NOT include any other analysis or explanations.

---

**Variance Detection Prompt**

Consider the following list of numbers representing a time series: $v_1, v_2, \ldots, v_n$. Some values may be missing (NaN). What is the variance of `channel` for this series, ignoring NaNs? Respond with only a single float rounded to 2 decimal places — no other text or numbers. Please DO NOT include any other analysis or explanations.

---

**Periodicity Detection Prompt**

Consider the following series: $v_1, v_2, \ldots, v_n$. Please investigate whether the series exhibits strong periodicity, ignoring any NaN values. If it does, respond with an integer value representing approximately how often strong periods occur in the series. If there is no evidence of strong periodicity, respond with the sequence length $n$. Do not include any other numbers in your response, whether in the form of intermediate calculations or steps. Remember you MUST return an INTEGER value or $n$. Please DO NOT include any other analysis or explanations.

---

**Trend Detection Prompt**

Consider the following series: $v_1, v_2, \ldots, v_n$. Please describe the average trend of the series, ignoring any NaN values. If the series is decreasing on average, respond with a value of -1. If it is increasing, respond with a value of 1. If there doesn't appear to be a strong trend in any direction, please respond with a value of 0. Note that wireless data can be noisy, so look at global changes to determine trend. Do not include any other numbers in your response, whether in the form of intermediate calculations or steps. ONLY RESPOND WITH -1, 0, or 1. Do NOT include any other analysis or explanations.

---

**Network QA Prompts**. These prompts are used to assess a model's network understanding capabilities. For each prompt, we provide the KPIs from a sample and ask the model to provide an answer to the question at hand.

---

**Network QA Prompt**

You are an AI assistant tasked with analyzing time series data for a wireless network. You will be provided with a time series dataset containing various metrics to analyze. The time series is sampled every 0.1 seconds.

Your goal is to answer the questions about the user's activity, location, network congestion, jammer presence, and motion status based on the provided time series data only.

The possible activities are: YouTube, Large file download, and Twitch. The possible zones are: Zone A (closest to the gNB), Zone B (middle), and Zone C (furthest). The possible congestion status is: Yes or No. The possible motion status is: Yes or No. The possible jammer presence is: Yes or No.

Time range : {$ts\_range[0]$} to {$ts\_range[-1]$}
{$metric_1$} : {$values_1$}
$\vdots$
{$metric_n$} : {$values_n$}

Now, answer the following questions:
Q1. What activity was the user engaged in?
Q2. Where was the user located?
Q3. Was the network congested?
Q4. Was the user in motion?
Q5. Was there a jammer present?

Do not include any reasoning, explanation, or commentary. You must return only the final answer using the format shown below, exactly as specified.

Respond with:
```
<activity>[your answer here]</activity>
<zone>[your answer here]</zone>
<congestion>[your answer here]</congestion>
<motion>[your answer here]</motion>
<jammer>[your answer here]</jammer>
```

---

## E. TelecomTS: An Example

We provide below a representative example from `TelecomTS` to illustrate the structure and content of the dataset.

**TelecomTS Sample with Structured Metadata**

```
{

    "start_time": "2025-07-07 00:07:21.600",
    "end_time": "2025-07-07 00:07:34.300",
    "sampling_rate_hz": 10,
    "KPIs": {
        "keys": [
            "RSRP", "DL_BLER", "DL_MCS", "UL_BLER", "UL_MCS",
            "UL_NPRB", "UL_SNR", "TX_Bytes", "RX_Bytes",
                ...
        ],
        "values": [
            [-106.0, -106.0, -106.0, ... ], [0.00132, ... ],
                ...
        ]
    },
    "anomalies": {
        "exists": true,
        "type": ["High Network Congestion (Gradual Buildup)"],
        "anomaly_duration": [{"start": 0, "end": 127}],
        "affected_kpis": ["UL_BLER", "TX_Bytes", ... ],
        "troubleshooting_tickets": ["High Network Congestion",
            "**Diagnose Summary:**\n- **Issue:** ... "]
    },
    "statistics": {
        "RSRP": {
            "mean": -106.0,
            "variance": 0.0,
            "trend": 0,
            "periodicity": 1
        },
         ...
    },
    "labels": {
        "zone": "B",
        "application": "Youtube",
        "mobility": "No",
        "congestion": "No",
        "anomaly_present": "Yes"
    },
    "QnA": {
        "network": [
            {"q": "Can we classify the user as moving?",
             "a": "The session involved a static user.",
             "reasoning": "First off, 5G NR configurations typically ... "},
                ...
        ],
        "timeseries": [
            {"q": "What is the var of RX_Bytes?", "a": 286.3,
             "reasoning": Okay, let's start by analyzing the ... "},
            {"q": "What is the avg value of RSRP?", "a": -106,
             "reasoning": Okay, let's start by looking at the ... "},
                ...
        ]
    },
    "description": "The radio link shows a steady downlink..."

}
```

We additionally provide an example reasoning trace generated for a root cause identification task, illustrating how the model systematically examines KPI values, identifies cross-channel patterns, and converges on a diagnosis.

---

Reasoning Trace Example

```
Okay, let's start by looking at the RSRP values. They're clustered between -123.7
and -122.3 dB with a mean of -122.6. That's pretty stable, so maybe the connection
isn't dropping. Now, the uplink SNR ranges from 12.1 to 21.5 dB with an average of
17.1. That's a decent range, but the downlink BLER is between 0.0033 and 0.019,
averaging 0.0088. That's higher than normal, indicating possible issues in downlink.

The uplink BLER is 0.0, which is perfect. So the problem might be on the downlink
side. Downlink MCS is constant at 7.0, which is lower than the uplink's mean of 9.3.
Lower MCS usually means lower data rates, which can be due to interference. Uplink
resource usage is low, but downlink PRB utilization has brief spikes up to 53%,
suggesting some congestion or interference during those spikes.

TX_Bytes have a big spike up to 222,860 Bytes, but RX_Bytes are low. That might mean
the network is trying to retransmit packets due to errors, which aligns with the BLER
increase. The high downlink BLER and low MCS despite good uplink metrics point towards
downlink interference. Since RSRP is stable, it's not a signal strength issue. Co-
channel interference often causes such patterns where signal quality degrades without
affecting RSRP much. So, the most likely cause is co-channel interference, especially
since the symptoms align with mild interference affecting downlink more than uplink.
```

## F. Reasoning Trace Synthesis and Validation

### F.1. Reasoning Trace Generation

To support training and evaluation of reasoning-capable and reinforcement learning–based models in a multi-modal setting, we curate structured reasoning traces for all network-level and anomaly Q&A instances, as these categories require cross-channel reasoning over coordinated KPI changes. Each trace consists of an explicit reasoning path produced by the model during generation. These traces are synthesized using a large instruction-tuned language model (Qwen3-32B) via rejection sampling over multiple candidate generations.

**Network-Level Descriptions Generation.**    To support reasoning trace synthesis, we first generate concise natural-language descriptions summarizing the network state associated with each time series sample. These descriptions are grounded in programmatically extracted signal features, including detected trends, jumps, statistical ranges, and anomaly-specific effects across KPIs. A strong instruction-tuned language model (GPT-o3) is used only as a surface realization step: it is prompted with structured summaries of KPI behavior, task-specific metadata, and anomaly symptoms derived from troubleshooting tickets, and tasked with reformulating this information into a coherent narrative. The model does not perform inference or labeling, but serves to standardize and contextualize the extracted signal-level information. These descriptions are subsequently used as auxiliary context during reasoning trace generation.

**Trace Generation Procedure.**    For each Q&A instance, we prompt the model to generate a final answer together with an explicit reasoning trace. To ensure faithfulness and reduce hallucinations, the model is conditioned on the ground-truth label and task-specific metadata during generation. This conditioning is used *only* to guide trace synthesis and is not exposed at inference time; the generated traces serve as supervision signals for downstream reasoning and reinforcement learning tasks.

For each instance, we sample multiple candidate generations and select the highest-quality trace using the scoring function described below.

**Scoring Function.**    Each candidate generation is assigned a composite score combining correctness, reasoning quality, and format compliance:

$$S = w_{\text{task}} \cdot s_{\text{task}} + w_{\text{reas}} \cdot s_{\text{reas}} + w_{\text{fmt}} \cdot s_{\text{fmt}}, \tag{1}$$

with $(w_{\text{task}}, w_{\text{reas}}, w_{\text{fmt}}) = (0.50, 0.30, 0.20)$ and each component bounded in $[0, 1]$. The **task score** $s_{\text{task}} \in \{0, 1\}$ is an exact-match indicator against the ground-truth answer. For anomaly bounds localization, $s_{\text{task}}$ is the temporal IoU between predicted and ground-truth intervals.

The **reasoning score** evaluates the diagnostic quality of the reasoning trace:

$$s_{\text{reas}} = \gamma \cdot s^+(y) + (1-\gamma) \cdot \left(1 - s^-(y)\right), \tag{2}$$

where $s^+(y)$ rewards the presence of patterns discriminative for the ground-truth class $y$ and $s^-(y)$ penalizes patterns associated with confusable classes that share overlapping KPI signatures, with $\gamma=0.6$. Patterns are task-specific regular expressions covering (i) KPI-specific numeric evidence (e.g., RSRP ranges for zone discrimination), (ii) temporal descriptors matching the fault mechanism (e.g., *linear decay* vs. *step change*), and (iii) fault-specific terminology. Each pattern carries a weight $w_p$, and scores are computed as $s^+(y)=\sum_{p\in\mathcal{P}_y^+} w_p \mathbf{1}[p \in \text{trace}] / \sum_p w_p$, and analogously for $s^-$. For anomaly bounds, the reasoning score additionally validates that referenced timesteps align with the ground-truth window and that the KPIs discussed match those affected by the anomaly type.

The **format score** $s_{\text{fmt}}$ verifies structural requirements:

$$s_{\text{fmt}} = \sum_{k=1}^{K} \phi_k \cdot \mathbf{1}[c_k], \tag{3}$$

where each indicator $c_k$ checks a format requirement—proper delimiters between reasoning and answer, substantial reasoning content (length thresholds), a well-formed final answer, and absence of metadata leakage—with weights $\phi_k$ summing to one.

**Metadata Leakage Prevention.** To prevent trivial reasoning and ensure that the generated traces remain suitable as supervision signals, we enforce strict metadata-leakage constraints during trace selection. We apply lexical blacklists and heavy score penalties to candidate generations that explicitly reference access to ground-truth labels, conditioning metadata, or privileged inputs (e.g., phrases such as "given the ground truth" or "as provided in the metadata"). Candidates violating these constraints are assigned near-zero format scores, ensuring that retained traces reflect well-structured, plausible reasoning rather than explicit label copying.

**Task-Specific Conditioning Metadata.** During trace synthesis, the model is provided with task-specific metadata to guide faithful and grounded reasoning. The conditioning information varies by task and is used exclusively during trace generation:

- **Root Cause Analysis:** Ground-truth anomaly type, network description, affected KPIs, and observed symptoms from the troubleshooting tickets.

- **Anomaly Detection:** Ground-truth anomaly presence and network description.

- **Anomaly Duration:** Ground-truth anomaly start and end indices, along with a randomly selected subset (25%) of KPI time series channels.

- **Network-Level Tasks (zone, activity, congestion, mobility):** Ground-truth label and network description.

**Final Trace Selection.** For each Q&A instance, we retain the highest-scoring candidate generation according to the composite scoring function. The resulting dataset contains aligned tuples of the *final answer* and the corresponding *reasoning trace*, which serve as supervision signals for reasoning-aware training, including supervised fine-tuning with explicit reasoning and reinforcement learning with verifiable rewards.

### F.2. Reasoning Trace Quality Assessment

To ensure the quality of the LLM-generated reasoning traces included in `TelecomTS`, we design a scoring function for automatic filtering and quality assurance without relying on human annotation. The scoring function evaluates each candidate trace on substantive criteria: verifying the inclusion of relevant symptoms, checking that the correct affected KPIs are identified, and ensuring consistency between described signal behaviors and the actual underlying measurements. For each sample, we generate multiple candidate traces and retain only the highest-scoring one, filtering out generations with factual inconsistencies or missing diagnostic information.

Figure 17 shows the score distributions for all generated candidates and for the best-per-sample selections. The candidate pool already exhibits high scores (mean 0.68, with the majority concentrated between 0.6 and 0.8), reflecting the rich metadata and contextual information provided to the LLM during generation. The best-per-sample selection shifts the distribution rightward (mean 0.75), with the interquartile range rising from [0.64, 0.70] to [0.69, 0.78] and the 90th percentile reaching 0.85. This confirms that retaining the top candidate per sample consistently yields well-grounded traces.

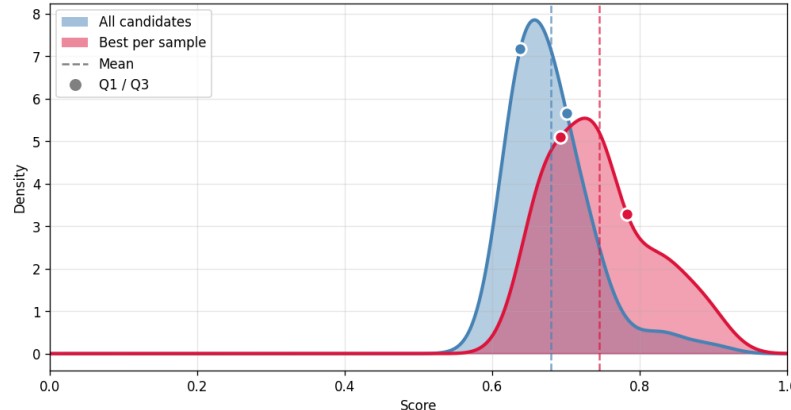

*Figure 17.* Score distributions of all generated trace candidates (blue) and best-per-sample selections (red).

To illustrate the quality of the resulting traces, we present a random example produced by a multi-modal model trained on `TelecomTS` for root cause identification. The trace demonstrates structured diagnostic reasoning: the model systematically examines individual KPIs, cross-references related indicators, rules out competing hypotheses, and converges on a diagnosis grounded in the observed signal patterns.

---

**Example reasoning trace: Root cause identification**

```
<think>
```
Okay, let's start by looking at the KPIs. The RSRP is weak but stable, which might suggest a signal strength issue, but BLER is zero, so maybe it's not a coverage problem.

▷ *Examining channel conditions*

The MCS values are fixed for UL and DL, which could indicate a lack of adaptability. The UL SNR is varying between 6.3 and 14.6 dB, which is a wide range. Even though the average is okay, the variability might imply interference.

▷ *Cross-referencing utilization and throughput*

The PRB utilization is very low in both directions, which is unexpected if the SNR is fluctuating. Since low utilization and stable BLER don't align with a coverage or capacity issue, maybe the interference is causing periods where the channel is usable but not stable. The low throughput and minimal packet activity support that the link isn't being used effectively despite the stable RSRP.

▷ *Converging on diagnosis*

This points towards interference on the same channel disrupting the signal stability. Therefore, the symptoms align with mild co-channel interference.

```
</think>
```
**Answer:** Co-channel interference (mild).

---

*Figure 18.* Example reasoning trace from a model trained on `TelecomTS`. Gray annotations highlight the reasoning stages; the trace text is the model's unedited output.

The trace exhibits several desirable properties: the model references specific KPI values (RSRP stability, BLER at zero, SNR range of 6.3–14.6 dB, low PRB utilization), identifies contradictions between indicators (fluctuating SNR with stable BLER and low utilization), eliminates competing explanations (coverage and capacity issues), and arrives at a diagnosis consistent with the observed evidence. These properties reflect the diagnostic structure encoded in the reasoning traces of `TelecomTS`, and their effectiveness is further validated by the performance gains observed when models are trained with reasoning (Section 4).

## G. Training Details

We train our models on `TelecomTS` using an 80–20 split between training and test data. To avoid label imbalance and potential bias, we ensure that the training subset is balanced across labels, both for anomaly detection and root cause analysis tasks. Forecasting is evaluated on both normal and anomalous sequences to provide a complete picture of performance under all conditions. For foundation models, only the classification or regression head is trained, while the backbone remains frozen.

For the early-fusion multi-modal model (Toto+Qwen3-4B), training proceeds in two stages. First, we perform time series encoder pre-alignment to bridge the temporal and textual modalities by training the Toto encoder and projection layer to predict network-level descriptions from KPI sequences, while keeping the language model backbone frozen. In the second stage, we perform joint end-to-end supervised fine-tuning across all supported tasks. The Toto encoder and projection layer remain fully trainable, while the language model backbone is fine-tuned using LoRA with rank $r = 16$, $\alpha = 16$, and a dropout rate of 0.05 applied to the query, key, value, output, and feed-forward projection layers.

For the reasoning-enabled variant (Toto+Qwen3-4B+Thinking), training follows the same pre-alignment stage, followed by a cold-start SFT phase for the same number of epochs where the model is trained on the curated reasoning traces (Appendix F.1) to produce structured reasoning followed by the final prediction.

Optimization across all stages is performed using Adam with $\beta = (0.9, 0.999)$, a learning rate of $10^{-3}$, weight decay of $10^{-4}$, a batch size of 32, and for 5 epochs. The training objective depends on the task: cross-entropy loss for classification, mean squared error for forecasting, and causal language modeling loss for the multi-modal setting.

## H. Challenges in Forecasting Observability Time Series

Forecasting in network observability settings poses distinct challenges that differ from those encountered in traditional time series benchmarks. Fig. 19 illustrates representative failure modes observed even for the highest-performing forecasting model (Informer) on our dataset.

**Delayed Peak Prediction.** Sudden bursts in KPIs such as throughput or buffer occupancy often arise from transient network events (e.g., congestion episodes or interference). These events are difficult to anticipate from past context alone, leading to temporally shifted predictions where peaks are detected with delay rather than proactively forecast.

**Inaccurate Magnitude Estimation.** Observability KPIs frequently exhibit heavy-tailed distributions and abrupt amplitude changes spanning several orders of magnitude. As a result, models may correctly identify the timing of an event but significantly underestimate or overestimate its severity, particularly for extreme values.

**Oscillatory and Non-Stationary Dynamics.** Several KPIs display irregular oscillations driven by feedback control mechanisms, scheduling policies, or adaptive protocols. These oscillatory patterns are often non-stationary and context-dependent, making them challenging to extrapolate using standard forecasting objectives that emphasize smoothness or trend continuity.

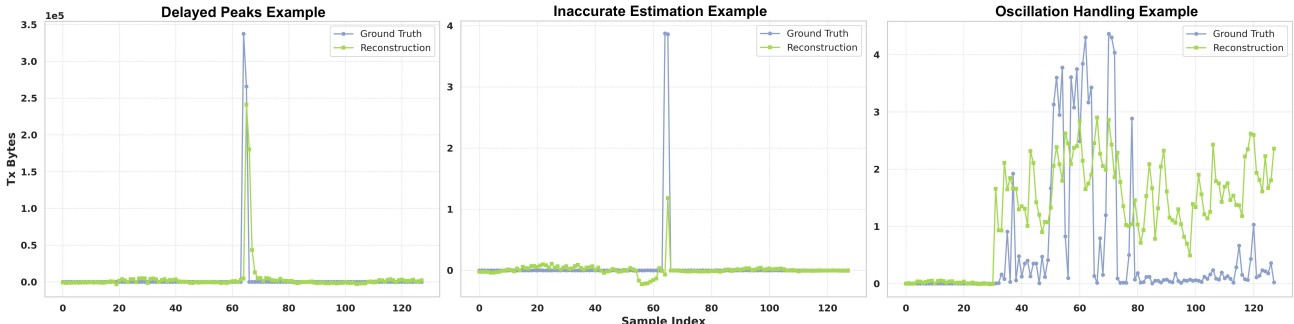

*Figure 19.* Forecasting results of the highest-performing model (Informer) highlight key challenges: (1) delayed peak predictions, (2) inaccurate magnitude estimation, and (3) difficulty in handling oscillatory patterns.

