# OpenReview forum: "TelecomTS: A Multi-Modal Observability Dataset for Time Series and Language Analysis"
_ICML.cc/2026/Conference — ICML 2026 regular_

### Official Review · Reviewer_4ZsN · 2026-02-27

**Soundness:** 2
**Presentation:** 2
**Significance:** 3
**Originality:** 3
**Overall Recommendation:** 3
**Confidence:** 4

**Summary:**

This paper introduces TelecomTS, a large-scale observability dataset derived from a 5G network. TelecomTS supports multiple downstream tasks, including anomaly detection, root-cause analysis, and multi-modal question-answering , complete with explicit reasoning paths.

**Compliance With Llm Reviewing Policy:**

Affirmed.

**Final Justification:**

The proposed dataset is valuable, but the experiments need to be supplemented and improved.

**Key Questions For Authors:**

1. The dataset excludes control-plane signaling and user-plane payload data, retaining only transport-layer headers. Could this omission oversimplify the "root-cause analysis" task by removing essential diagnostic information?
2. Evaluating multimodal models using data generated by Large Language Models (LLMs) inevitably introduces the source model's inherent biases. How does the study address or mitigate this issue?
3.It is suggested to add a quantitative evaluation of the generated "reasoning traces" (e.g., human blind review or LLM-as-a-judge). Without such metrics, it is difficult to confirm whether the models have genuinely learned logical diagnosis or simply overfit to shortcut patterns.

**Limitations:**

I do not see limitations are discussed.

**Strengths And Weaknesses:**

Strengths:
1.Unlike existing alternatives (like BOOM) that anonymize and normalize data , TelecomTS provides fully de-anonymized covariates and preserves absolute scale, which the authors prove is vital for downstream observability tasks.
2.By testing these across practical tasks like anomaly duration, root cause analysis, and multi-modal Q&A , the experiments effectively expose the severe limitations and false-positive biases of current architectures when handling naturally erratic data.
Weaknesses:
1.Although the dataset includes real anomalies generated via an adversarial jammer , a significant portion of the anomalies are synthetically generated by applying mathematical transformations to normal data. Furthermore, the troubleshooting tickets and reasoning traces are synthesized using LLMs (GPT-4.1 and Qwen3-32B).

---

> ### Author Rebuttal · Authors · 2026-03-31
>
> We thank the reviewer for their feedback and for recognizing the significance and originality of TelecomTS as a large-scale observability dataset. We address each point below:
>
> **1. Exclusion of control-plane and payload data.** TelecomTS does, in fact, include RAN control-layer signals such as RSRP, SNR, MCS, and BLER — key indicators of radio link quality and scheduling behavior. What is excluded is user-plane payload data, which is rarely accessible to network operators due to privacy constraints and is not typically part of operational diagnosis pipelines [1, 2]. To that end, root-cause analysis therefore requires genuine multi-KPI, cross-channel reasoning: anomalies manifest through coordinated changes across multiple signals (e.g., simultaneous BLER↑, MCS↓, PRB spikes), as illustrated in Table 9 (Appendix G), which shows closely related anomaly types sharing highly overlapping KPI signatures that demand strong temporal reasoning to differentiate.
>
> **2. LLM-generated content and evaluation bias.** We begin by noting that all KPI measurements, network configurations, and anomaly injections involve no LLM component, as the raw time-series data is entirely hardware-collected. LLM-generated content is used exclusively in two components: (1) troubleshooting tickets and (2) reasoning traces. For troubleshooting tickets, the LLM is provided with explicit ground-truth information, including the anomaly cause, start and end times, affected KPIs, and behavioral signatures—and is tasked with synthesizing a structured ticket, which is a simple synthesis task. Human validation is limited to reviewing sampled tickets to ensure that symptoms, affected KPIs, and conclusions are accurately reflected. For reasoning traces, we design a scoring function to enable automatic filtering and ensure quality without relying on human annotation. Specifically, the scoring verifies the inclusion of relevant symptoms, checks that the number of affected KPIs is correctly identified, and ensures consistency between described symptoms and underlying signals. We generate multiple candidate traces per sample and retain the most well-grounded generation, filtering out those with inconsistencies (see Appendix B.3). We will further clarify this process in the camera-ready version.
>
> **3. Realism of synthetic anomalies.** Our dataset includes two types of anomalies: (1) real anomalies generated using a jammer during controlled experiments, and (2) synthetic anomalies designed to mimic real-world conditions. Given the rarity of anomalies, the synthetic portion aims to expand the coverage of scenarios observed in networks beyond jamming. To ensure realism, the synthetic anomalies are grounded in established telecom literature and replicate characteristic behaviors reported in prior work. For example, co-channel interference anomaly is modeled to degrade SNR, BLER, and PRB utilization as reported in [3]; congestion and buffer overflow exhibit PRB saturation and declining throughput [4]; Doppler-induced channel aging leads to lower MCS and elevated BLER [5]; and ping-pong handover results in oscillatory signal quality and increased packet loss [6].
>
> [1] Xu et al., Understanding Operational 5G: A First Measurement Study, ACM SIGCOMM 2020
>
> [2] Hasan et al., Root Cause Analysis of Anomalies in 5G RAN Using Graph Neural Network and Transformer, arXiv:2406.15638
>
> [3] Tusha et al., A Comprehensive Analysis of Secondary Coexistence in a Real-World CBRS Deployment, arXiv:2402.05226
>
> [4] Koivisto et al., Congestion Analysis of Transport Layer in a Multicell 5G DL Communication System, MDPI Sensors, 2023
>
> [5] Díaz-Ruiz et al., Deep Learning-Based CSI Prediction Framework for Channel Aging Mitigation in TDD 5G Systems, arXiv:2510.24400
>
> [6] Tsai et al., Using Fuzzy Logic to Reduce Ping-Pong Handover Effects in LTE Networks, Soft Computing, 2016

---

> > ### Author Rebuttal · Reviewer_4ZsN · 2026-04-02
> >
> > Thank you for the response. However, no specific settings or results were provided for the quantitative assessment of LLM deviations; moreover, since there are both real anomalies and synthetic anomalies in the dataset, why weren't they compared, and even an experiment designed to verify whether the model trained on purely synthetic anomalies can generalize to real data? I have decided to maintain the original score.

---

> > > ### Author Response · Authors · 2026-04-02
> > >
> > > We thank the reviewer for the follow-up. We address both points below.
> > >
> > > 1. **Quantitative assessment of LLM-generated reasoning traces.** To address this, we designed a scoring function (Appendix B.3) that evaluates reasoning traces on specific criteria: verifying the inclusion of relevant symptoms, checking that the correct number of affected KPIs is identified, and ensuring consistency between described signal behaviors and the actual underlying measurements. Following the reviewer's suggestion (and that of reviewer **pqxm**), we provide summary statistics of the score distribution used to assess the validity of LLM-generated reasoning traces. As shown, the resulting score distribution is highly concentrated around 0.8, indicating consistently high-quality candidates with low variance. This is attributable to the rich metadata and contextual information provided to the LLM during generation of these traces (as we detailed in Appendix B.3). Moreover, for each sample, we retain only the highest-scoring candidate among multiple generated reasoning traces to ensure the quality of the final traces. We will include this analysis in the camera-ready version.
> > >
> > >     | Metric | Value |
> > >     | --- | --- |
> > >     | Mean ± Std | 0.848 ± 0.085 |
> > >     | Q1 / Median / Q3 | 0.8 / 0.8 / 1.0 |
> > >     | 95th percentile | 1.0 |
> > >
> > > 2. **Real vs. synthetic anomaly comparison.** We clarify that real and synthetic anomalies in TelecomTS serve complementary, not interchangeable, roles. The real anomalies originate from adversarial jamming, which affects a specific subset of KPIs. The synthetic anomalies, by contrast, are designed to cover fundamentally different fault types (co-channel interference, congestion, ping-pong handover, etc.), each manifesting through unique cross-channel signatures. These are distinct phenomena with different affected KPI patterns. Therefore, a model trained on jamming is not expected to generalize effectively to congestion faults, as the underlying signatures are distinct, similar to a classifier trained to recognize cats failing to recognize dolphins for example. Hence, the purpose of synthetic anomalies is to **expand** the coverage of fault types beyond what a single real-world anomaly source can provide, not to serve as a proxy for real anomalies.
> > >
> > > We hope our responses have addressed the reviewer's concerns and improved the reviewer’s overall assessment of the paper.

---

### Official Review · Reviewer_FmPZ · 2026-03-08

**Soundness:** 3
**Presentation:** 3
**Significance:** 3
**Originality:** 3
**Overall Recommendation:** 4
**Confidence:** 3

**Summary:**

This paper introduces TelecomTS, a multi-modal observability dataset for telecom time series, with de-anonymized covariates, anomaly labels, and QA-style tasks. The paper also benchmarks a range of time-series, language, and multi-modal models, and argues that realistic observability data is harder than standard smooth benchmarks, especially when absolute scale information matters.

**Compliance With Llm Reviewing Policy:**

Affirmed.

**Final Justification:**

My concerns have been well resolved. I will keep my positive score.

**Key Questions For Authors:**

1. Can the authors better validate that the synthetic anomalies reflect realistic telecom failures? A stronger validation would increase my confidence in the benchmark.
2. How reliable are the LLM-generated reasoning traces after filtering? Some manual audit would be helpful.

**Limitations:**

see weakness

**Strengths And Weaknesses:**

Strengths:

1. The paper introduces a useful multi-modal observability dataset for telecom time series.
2. The benchmark covers several practical tasks and a broad set of baselines.
3. The results suggest that realistic observability data remains challenging for current models.

Weaknesses:

1. The benchmark depends quite a bit on synthetic anomaly injection and GPT-4.1-generated tickets/reasoning traces. I am not yet fully convinced that these components are realistic enough for a benchmark meant to reflect telecom observability.
2. The scale-preservation claim is promising, but the empirical support is still limited, since the main direct evidence is a relatively small ablation.
3. A few evaluation decisions need more explanation, especially the 1,000-sample anomaly detection subset with an 80/20 class ratio and the use of selected KPI slices in analysis. Figure 1 is also hard to read.

---

> ### Author Rebuttal · Authors · 2026-03-31
>
> We thank the reviewer for their constructive feedback. We address each point below:
>
> **1. Realism of synthetic anomalies.** Our dataset includes two types of anomalies: (1) real anomalies generated using a jammer during controlled experiments, and (2) synthetic anomalies designed to mimic real-world conditions. Given the rarity of anomalies, the synthetic portion aims to expand the coverage of scenarios observed in networks beyond jamming. To ensure realism, the synthetic anomalies are grounded in established telecom literature and replicate characteristic behaviors reported in prior work. For example, co-channel interference anomaly is modeled to degrade SNR, BLER, and PRB utilization as reported in [1]; congestion and buffer overflow exhibit PRB saturation and declining throughput [2]; Doppler-induced channel aging leads to lower MCS and elevated BLER [3]; and ping-pong handover results in oscillatory signal quality and increased packet loss [4].
>
> **2. LLM-generated content and validation.** We begin by noting that all KPI measurements, network configurations, and anomaly injections involve no LLM component, as the raw time-series data is entirely hardware-collected. LLM-generated content is used exclusively in two components: (1) troubleshooting tickets and (2) reasoning traces.For troubleshooting tickets, the LLM is provided with explicit ground-truth information, including the anomaly cause, start and end times, affected KPIs, and behavioral signatures—and is tasked with synthesizing a structured ticket, which is a simple synthesis task. Human validation is limited to reviewing sampled tickets to ensure that symptoms, affected KPIs, and conclusions are accurately reflected. For reasoning traces, we design a scoring function to enable automatic filtering and ensure quality without relying on human annotation. Specifically, the scoring verifies the inclusion of relevant symptoms, checks that the number of affected KPIs is correctly identified, and ensures consistency between described symptoms and underlying signals. We generate multiple candidate traces per sample and retain the most well-grounded generation, filtering out those with inconsistencies (see Appendix B.3). We will further clarify this process in the camera-ready version.
>
> **3. Scale preservation.** Absolute scale encodes critical physical operating regimes directly tied to downstream tasks; for example, user location and signal quality depend on absolute received power values, not relative variations alone. Following the reviewer's suggestion, we conducted an ablation across five model architectures, comparing each against its non-scaled version. Scale-aware models consistently outperform their ablated versions, with the most pronounced drops in RCA and anomaly detection. This is further supported by Tables 2–5, where normalized models such as Autoformer consistently underperform, and Mantis outperforms Mantis (w/o scaling). The ablation will be included in the camera-ready version.
>
> ||Root Cause (w→w/o scale)|Anom Detection F1 (w→w/o scale)|Zone (w→w/o scale)|Activity (w→w/o scale)|Cong (w→w/o scale)|Motion (w→w/o scale)|
> |-|-|-|-|-|-|-|
> |TimesNet|62.4→40.1|87.1→80.4|88.7→74.3|99.5→99.5|99.0→98.3|91.8→96.7|
> |Non-Stationary Transformer|48.2→38.5|87.4→80.9|88.1→66.2|99.4→99.3|97.9→96.9|96.2→93.4|
> |Mantis|79.4→72.1|96.4→84.2|95.8→82.0|100.0→99.6|99.5→96.5|99.2→98.9|
> |Toto|72.9→61.1|85.3→80.0|93.4→83.5|99.8→99.6|99.1→95.1|97.5→98.5|
> |Chronos|38.5→28.3|75.3→70.3|67.5→51.9|97.5→92.6|94.0→80.4|94.6→81.9|
>
> **4. Evaluation decisions.** We balance training data to avoid class imbalance bias. At evaluation, we adopt an 80/20 normal/anomaly ratio, consistent with anomaly prevalence ranges used in established anomaly detection benchmarks [5, 6], and chosen to make reported metrics more representative of realistic deployment conditions where network anomalies are rare events. The 1,000-sample subset balances computational cost across multiple large models while maintaining representative and stable evaluation. In all experiments, we include all KPI channels in the input. We will clarify these decisions and improve Figure 1's readability in the camera-ready version.
>
> [1] Tusha et al., A Comprehensive Analysis of Secondary Coexistence in a Real-World CBRS Deployment, arXiv:2402.05226
>
> [2] Koivisto et al., Congestion Analysis of Transport Layer in a Multicell 5G DL Communication System, MDPI Sensors, 2023
>
> [3] Díaz-Riz et al., Deep Learning-Based CSI Prediction Framework for Channel Aging Mitigation in TDD 5G Systems, arXiv:2510.24400
>
> [4] Tsai et al., Using Fuzzy Logic to Reduce Ping-Pong Handover Effects in LTE Networks, Soft Computing, 2016
>
> [5] Han et al., ADBench: Anomaly Detection Benchmark, arXiv:2206.09426
>
> [6] Xu et al., Anomaly Transformer: Time Series Anomaly Detection with Association Discrepancy, arXiv:2110.02642

---

> > ### Author Rebuttal · Reviewer_FmPZ · 2026-04-04
> >
> > Thanks for the authors' response. My concerns have been well resolved. I will keep my positive score.

---

### Official Review · Reviewer_pqxm · 2026-03-09

**Soundness:** 3
**Presentation:** 3
**Significance:** 2
**Originality:** 2
**Overall Recommendation:** 5
**Confidence:** 4

**Summary:**

TelecomTS is a large-scale observability dataset collected from a 5G telecommunications network, motivated by the scarcity of publicly available, de-anonymized observability data for time series research. The authors position TelecomTS against existing observability datasets by providing visibility into covariate identities and scale, along with a selection of multi-modal downstream tasks including anomaly detection, root cause analysis, forecasting, and question-answering. Anomalies are sourced from both real adversarial jamming and different types of synthetic perturbations grounded in telecommunications literature. Benchmarks across the aforementioned tasks indicate that existing approaches struggle with the dynamics of observability data, while models that encode scale information perform better.

**Compliance With Llm Reviewing Policy:**

Affirmed.

**Final Justification:**

I increased my score to 5 following the initial rebuttal to reflect the utility and relevance of such work, which has the possibility to steer researchers towards new open problems and questions that may not be well-represented in existing benchmark datasets.

**Key Questions For Authors:**

1. As I mentioned in an earlier section of the review, does forecasting evaluation include samples with injected synthetic anomalies? If so, results on anomaly-free samples would help disentangle forecasting difficulty from the effect of synthetic perturbations.

2. The human validation of LLM-generated content is mentioned but only briefly. Could you provide more details on this?

**Limitations:**

Unless I missed it, the paper does not include a dedicated limitations section.

**Strengths And Weaknesses:**

**Strengths**

- Researchers in the time series community regularly wish for industry partners holding large proprietary datasets to surface real-world data that reveals new problems beyond the standard benchmarks that have traditionally characterized the field. This dataset offers a meaningful step in that direction as the high degree of heterogeneity reflects diversity that practitioners often encounter in real environments.

- I appreciated the thoroughness of the data collection documentation, which sets a good standard for reproducibility in dataset papers.

- The multi-task design and benchmarking effort is substantial, covering multiple model families across different task types. The ablation studies are insightful and the failure case analysis effectively communicates the core challenges posed by the dataset.

**Weaknesses**

- The dataset is collected from a base station in a controlled lab. While the authors explain this is necessary as telecommunications data is proprietary, the implications for generalizability are not clear. I would like to understand more how much of the erratic/bursty behaviour is representative of production networks versus an artifact of the lab's set-up. A discussion of these limitations would help calibrate the conclusions.

- A substantial portion of the dataset relies on LLM-generated content which sits in some tension with the motivation of surfacing real-world observability data. I don't think this necessarily undermines the dataset's value, but I would appreciate more detail on the human/automated validation process.

- While the multi-task design is a strength overall, I found the individual tasks unevenly convincing. The time series Q&A component appears to reduce to four deterministic computations applied per KPI, which seems to test numerical computation more than genuine temporal reasoning. The forecasting evaluation acknowledges that MAE is inflated by stable segments, but it is unclear whether samples with injected synthetic anomalies are included in the evaluation. The anomaly detection and root cause analysis tasks feel more naturally suited to this data.

- A more systematic evaluation across multiple architectures would make the claim on the importance of scale information more convincing. I would also be curious to know whether the model rankings observed here differ meaningfully from those reported against BOOM or other observability benchmarks, as this would help clarify what TelecomTS uniquely reveals.

---

> ### Author Rebuttal · Authors · 2026-03-31
>
> We thank the reviewer for the constructive feedback and for recognizing the dataset's value for capturing real-world network observability. We address each point below:
>
> **1. Generalizability of lab-collected data.** While TelecomTS is collected from a controlled 5G testbed, our data collection deliberately varied network configurations, transmission power levels, traffic patterns, and congestion levels to ensure generalization. The fundamental phenomena captured, such as interference, congestion, and scheduling dynamics, are invariant across deployments, making models naturally generalizable to real-world networks. Crucially, the burstiness and heavy-tailed variability observed in TelecomTS are well-established intrinsic properties of radio network behavior, driven by the volatile wireless channel, scheduling dynamics, and retransmission mechanisms, as documented in production network measurement studies [1, 2]. Please note that operator network deployments are proprietary and inaccessible, making TelecomTS particularly valuable for capturing aspects of the observability domain.
>
> **2. LLM-generated content.** We note that all KPI measurements, network configurations, and anomaly injections involve no LLM component, as the raw time-series data is entirely hardware-collected. LLM-generated content is used exclusively in two components: (1) troubleshooting tickets and (2) reasoning traces. For troubleshooting tickets, the LLM is provided with explicit ground-truth information, including the anomaly cause, start/end times, affected KPIs, and behavioral signatures, and is tasked with synthesizing a structured ticket, which is a simple synthesis task. Human experts reviewed sampled tickets to ensure that symptoms, affected KPIs, and conclusions are accurately reflected. For reasoning traces, we design a scoring function for automatic filtering and quality assurance without relying on human annotation. Specifically, the scoring verifies symptom inclusion, correct affected KPI identification, and consistency between described symptoms and underlying signals. We generate multiple traces per sample and retain the best-grounded generation, filtering out inconsistencies (Appendix B3). We will further clarify this process in the camera-ready version.
>
> **3. Task design.** Our philosophy behind TelecomTS is to cover a spectrum of temporal reasoning difficulty. At the simpler end, *time series QA* includes basic statistical queries, thus testing whether a model can extract and interpret numerical properties from raw KPIs. However, our dataset also includes more advanced QAs requiring deeper reasoning. In fact, *network-level* and *anomaly QA* require reasoning over coordinated changes across multiple KPIs. For example, diagnosing a fault requires recognizing that BLER↑, MCS↓, and PRB spikes co-occur and are interdependent. Table 9 illustrates this by showing overlapping KPI signatures across fault types, which require cross-channel reasoning for the model to distinguish among.
>
> **4. Forecasting evaluation:** We clarify that forecasting is evaluated on both normal and anomalous sequences to provide a complete picture of performance under all conditions.
>
> **5. Scale information.** Absolute scale encodes critical physical operating regimes directly tied to downstream tasks; for example, user location and signal quality depend on absolute power values, not relative variations alone. Following the reviewer's suggestion, we conducted an ablation across five model architectures, comparing each against its non-scaled version. Scale-aware models consistently outperform their ablated versions, with the most pronounced drops in RCA and anomaly detection. This is further supported by Tables 2–5, where normalized models like Autoformer underperform, and Mantis outperforms Mantis (w/o scaling). The ablation will be included in the camera-ready version.
> ||Root Cause (w→w/o scale)|Anom Detection F1 (w→w/o scale)|Zone (w→w/o scale)|Activity (w→w/o scale)|Cong (w→w/o scale)|Motion (w→w/o scale)|
> |-|-|-|-|-|-|-|
> |TimesNet|62.4→40.1|87.1→80.4|88.7→74.3|99.5→99.5|99.0→98.3|91.8→96.7|
> |Non-Stationary Transformer|48.2→38.5|87.4→80.9|88.1→66.2|99.4→99.3|97.9→96.9|96.2→93.4|
> |Mantis|79.4→72.1|96.4→84.2|95.8→82.0|100.0→99.6|99.5→96.5|99.2→98.9|
> |Toto|72.9→61.1|85.3→80.0|93.4→83.5|99.8→99.6|99.1→95.1|97.5→98.5|
> |Chronos|38.5→28.3|75.3→70.3|67.5→51.9|97.5→92.6|94.0→80.4|94.6→81.9|
>
> **6. Model rankings.** The ranking of models reported in BOOM’s paper on forecasting is Toto>Moirai>Chronos>DLinear. Interestingly, we observe the same ordering in TelecomTS across multiple tasks, including RCA and zone prediction. This consistency suggests TelecomTS rankings are aligned with broader observability benchmarks. We will include this in the camera-ready version.
>
> [1] Xu et al., Understanding Operational 5G: A First Measurement Study, SIGCOMM 2020
>
> [2] Narayanan et al., A Variegated Look at 5G in the Wild: Performance, Power, and QoE Implications, SIGCOMM 2021

---

> > ### Author Rebuttal · Reviewer_pqxm · 2026-04-01
> >
> > Thank you for the detailed responses and performing the ablation study - I think this makes the significance of sharing unnormalized observations much clearer. Regarding the LLM validation process, I would still suggest reporting some form of inter-annotator agreement or summary statistics on LLM-generated content as this was a recurring concern across reviews.
> >
> > I would also like to shift my assessment of the paper to being more positive. Collecting the data and preparing the associated benchmarks is a commendable effort, and this kind of work sometimes benefits the research community more durably than many incremental modelling improvements. I am raising my score to 5 to better reflect this, while encouraging the authors to carefully address all questions arising in the reviews that may similarly emerge from future researchers or practitioners using this dataset.

---

> > > ### Author Response · Authors · 2026-04-02
> > >
> > > We thank the reviewer for engaging with our rebuttal and we appreciate the shift towards a more positive assessment of our paper. Following the reviewer’s suggestion, we provide summary statistics of the score distribution (defined in Appendix B3) used to assess the validity of LLM-generated reasoning traces. As shown in the table below, the distribution is highly concentrated, with most traces clustered around 0.8, indicating consistently high-quality candidates with low variance. This can be attributed to the metadata and additional contextual information provided to the LLM, which facilitate the generation of high-quality reasoning traces, as detailed in Appendix B.3. Moreover, for each sample, we retain only the highest-scoring candidate among multiple generated reasoning traces to ensure the quality of the final traces. We will include this analysis in the camera-ready version.
> > >
> > > | Metric | Value |
> > > | --- | --- |
> > > | Mean ± Std | 0.848 ± 0.085 |
> > > | Q1 / Median / Q3 | 0.8 / 0.8 / 1.0 |
> > > | 95th percentile | 1.0 |

---

### Official Review · Reviewer_ueki · 2026-03-15

**Soundness:** 2
**Presentation:** 2
**Significance:** 2
**Originality:** 2
**Overall Recommendation:** 4
**Confidence:** 4

**Summary:**

This paper introduces TelecomTS, a large-scale multi-modal dataset for studying time-series analysis and language reasoning in observability systems, focusing on telecommunications networks. Observability data differs from traditional time-series datasets because it is highly stochastic, zero-inflated, and exhibits abrupt bursts rather than smooth temporal patterns, making common forecasting benchmarks insufficient.

The dataset is collected from a controlled 5G network testbed, producing over 1 million observations across 18 KPIs with heterogeneous covariates and preserved scale information. It includes both real anomalies (e.g., signal jamming) and synthetic anomalies designed to mimic realistic network failures, and supports tasks such as anomaly detection, root-cause analysis, forecasting, and time-series question answering.

Benchmark experiments with language models, reasoning models, and time-series models show that current approaches struggle with the noisy and bursty dynamics of observability data, highlighting the need for scale-aware time-series models tailored to real-world monitoring systems.

**Compliance With Llm Reviewing Policy:**

Affirmed.

**Key Questions For Authors:**

A large portion of the anomalies are synthetically generated. Could the authors provide additional evidence that these synthetic anomalies realistically reflect real-world telecom failures (e.g., comparisons with real incident traces)? If the synthetic generation process closely matches real operational anomalies, it would strengthen confidence in the dataset’s practical relevance.

The dataset is collected from a single controlled 5G testbed. How well do the authors expect models trained on TelecomTS to generalize to real telecom deployments with different hardware, traffic patterns, or network configurations? Evidence of cross-network robustness would increase the dataset’s broader applicability.

The paper emphasizes the importance of preserving absolute scale information in KPIs. Could the authors provide a more systematic analysis of how scale normalization affects model performance across tasks? A clearer quantitative study would help validate this design choice.

Many Q&A questions appear to be derived from metadata or simple statistical properties of the time series. How do the authors ensure that the QA tasks require genuine temporal reasoning rather than simple statistical extraction? If the tasks involve deeper reasoning about system behavior, it would strengthen the multimodal evaluation claim.

For language models and reasoning models, the prompts provide varying levels of context (e.g., anomaly descriptions or wireless domain hints). Could the authors clarify how prompt design affects performance and whether different prompt strategies were tested to ensure fair comparisons?

**Limitations:**

yes

**Strengths And Weaknesses:**

Pros

The paper introduces a dataset from the observability domain (telecom networks), which is rarely available publicly due to proprietary constraints and reflects challenging real-world characteristics such as bursty and stochastic dynamics.

TelecomTS contains over 1M observations across 18 KPIs with heterogeneous numerical and categorical covariates and preserves absolute scale information, enabling a wider range of tasks beyond standard forecasting.

The dataset supports anomaly detection, anomaly localization, root-cause analysis, forecasting, and time-series QA, enabling evaluation of both time-series models and multimodal reasoning systems.

Cons

Although real anomalies are included, a significant portion of anomalies are synthetically generated, which may not fully capture the complexity of real-world failures.

The dataset is collected from a single controlled 5G testbed, which may limit diversity across different network infrastructures or operational environments.

The question-answering tasks are largely template-based and derived from metadata or simple statistics, which may limit the difficulty and realism of the multimodal reasoning evaluation.

---

> ### Author Rebuttal · Authors · 2026-03-31
>
> We thank the reviewer for their thoughtful feedback and for recognizing TelecomTS as a rare public dataset capturing real-world observability domains. We address each of your concerns below:
>
> **1. Realism of synthetic anomalies.** Our dataset includes two types of anomalies: (1) real anomalies generated using a jammer during controlled experiments, and (2) synthetic anomalies designed to mimic real-world conditions. Given the rarity of anomalies, the synthetic portion aims to expand the coverage of scenarios observed in networks beyond jamming. To ensure realism, the synthetic anomalies are grounded in established telecom literature and replicate characteristic behaviors reported in prior work. For example, co-channel interference anomaly is modeled to lower SNR, BLER, and PRB utilization as reported in [1]; congestion and buffer overflow exhibit PRB saturation and declining throughput [2]; Doppler-induced channel aging leads to lower MCS and elevated BLER [3]; and ping-pong handover results in oscillatory signal quality and increased packet loss [4].
>
> **2. Generalization across deployments.** While TelecomTS is collected from a controlled 5G testbed, our data collection deliberately varied network configurations, transmission power levels, traffic patterns, and congestion levels to ensure generalization. Moreover, the fundamental phenomena captured, such as interference and congestion, are invariant across deployments, making our dataset naturally generalizable to real-world networks. Please note that operator network deployments are proprietary and inaccessible, making TelecomTS particularly valuable for capturing aspects of the observability domain.
>
> **3. Scale influence.** Absolute scale encodes critical operating regimes directly tied to downstream tasks; for example, user location and signal quality depend on absolute received power values, not relative variations alone. Following the reviewer's suggestion, we conducted an ablation across five architectures, comparing each against its non-scaled version. Scale-aware models consistently outperform their ablated versions, with the most pronounced drops in RCA and anomaly detection. This is further supported by Tables 2–5, where normalized models like Autoformer underperform, and Mantis outperforms Mantis (w/o scaling). The ablation will be included in the camera-ready version.
> ||Root Cause (w→w/o scale)|Anom Detection F1 (w→w/o scale)|Zone (w→w/o scale)|Activity (w→w/o scale)|Cong (w→w/o scale)|Motion (w→w/o scale)|
> |-|-|-|-|-|-|-|
> |TimesNet|62.4→40.1|87.1→80.4|88.7→74.3|99.5→99.5|99.0→98.3|91.8→96.7|
> |Non-Stationary Transformer|48.2→38.5|87.4→80.9|88.1→66.2|99.4→99.3|97.9→96.9|96.2→93.4|
> |Mantis|79.4→72.1|96.4→84.2|95.8→82.0|100.0→99.6|99.5→96.5|99.2→98.9|
> |Toto|72.9→61.1|85.3→80.0|93.4→83.5|99.8→99.6|99.1→95.1|97.5→98.5|
> |Chronos|38.5→28.3|75.3→70.3|67.5→51.9|97.5→92.6|94.0→80.4|94.6→81.9|
>
> **4. QA tasks and temporal reasoning.** Our philosophy behind TelecomTS is to cover a spectrum of temporal reasoning difficulty. At the simpler end, *time series QA* includes basic statistical queries, thus testing whether a model can extract and interpret numerical properties from raw KPI signals. However, our dataset also includes more advanced QAs requiring deeper temporal reasoning. In fact, *network-level* and *anomaly QA* require reasoning over coordinated changes across multiple KPIs. For example, diagnosing a fault requires recognizing BLER↑, MCS↓, and PRB spikes are co-occurring and interdependent. Table 9 illustrates this by showing overlapping KPI signatures across fault types, which require cross-channel reasoning for the model to distinguish among.
>
> **5. Prompt design and fairness.** We evaluated multiple prompt strategies and found that prompt variations have a limited impact relative to the data's intrinsic difficulty (Appendix C). Formatting inconsistencies in the conclusion block were rare (~3%) and handled via regeneration, confirming robustness to prompt design.
>
> **6. Template-based QA.** While QAs follow a consistent answer structure for parsing purposes, the underlying reasoning required to arrive at the answer is non-trivial. Much like math problems, where the answer is a number but the steps to reach it are complex, the same applies here. Consider root cause: although the expected output is a class label, arriving at it requires analyzing temporal KPI trajectories, identifying multi-channel deviations, and distinguishing failure types with overlapping signatures.
>
> [1] Tusha et al., A Comprehensive Analysis of Secondary Coexistence in a Real-World CBRS Deployment, arXiv:2402.05226
>
> [2] Koivisto et al., Congestion Analysis of Transport Layer in a Multicell 5G DL Communication System, MDPI Sensors, 2023
>
> [3] Díaz-Riz et al., Deep Learning-Based CSI Prediction Framework for Channel Aging Mitigation in TDD 5G Systems, arXiv:2510.24400
>
> [4] Tsai et al., Using Fuzzy Logic to Reduce Ping-Pong Handover Effects in LTE Networks, Soft Computing, 2016

---

### Decision · Program_Chairs · 2026-04-30

**Decision:**

Accept (regular)

**Comment:**

This paper introduces a valuable multi-modal observability dataset from a 5G network. The reviewers generally agree that it fills an important gap by providing rare public data with de-anonymized covariates, anomaly labels, and multiple realistic tasks. The main strengths are the practical relevance of the dataset. But the main concerns were the controlled single-testbed setting, the reliance on synthetic anomalies, the use of LLM-generated tickets and reasoning traces, etc. The authors addressed many of these points well in the rebuttal by clarifying the role of LLM-generated content, providing validation details and score statistics for reasoning traces, and adding stronger scale ablations, which resolved the concerns of two reviewers and led one to raise their score. One reviewer remained only partially satisfied, mainly due to limited quantitative validation of LLM-generated content and the lack of stronger comparison between real and synthetic anomalies.